# A Comprehensive Review of the Current and Future Role of the Microbiome in Pancreatic Ductal Adenocarcinoma

**DOI:** 10.3390/cancers14041020

**Published:** 2022-02-17

**Authors:** Nabeel Merali, Tarak Chouari, Kayani Kayani, Charles J. Rayner, José I. Jiménez, Jonathan Krell, Elisa Giovannetti, Izhar Bagwan, Kate Relph, Timothy A. Rockall, Tony Dhillon, Hardev Pandha, Nicola E. Annels, Adam E. Frampton

**Affiliations:** 1Minimal Access Therapy Training Unit (MATTU), Leggett Building, University of Surrey, Daphne Jackson Road, Guildford GU2 7WG, UK; n.merali@nhs.net (N.M.); t.rockall@nhs.net (T.A.R.); 2Department of Hepato-Pancreato-Biliary (HPB) Surgery, Royal Surrey County Hospital, Egerton Road, Guildford GU2 7XX, UK; t.chouari@nhs.net (T.C.); kayani@doctors.org.uk (K.K.); c.rayner@surrey.ac.uk (C.J.R.); 3Targeted Cancer Therapy Unit, Department of Clinical and Experimental Medicine, Faculty of Health and Medical Science, University of Surrey, Guildford GU2 7WG, UK; izhar.bagwan@nhs.net (I.B.); k.relph@surrey.ac.uk (K.R.); tony.dhillon@nhs.net (T.D.); hardev.pandha@nhs.net (H.P.); n.annels@surrey.ac.uk (N.E.A.); 4Institute of Cancer and Genomic Sciences, College of Medical and Dental Sciences, University of Birmingham, Edgbaston, Birmingham B15 2TT, UK; 5Department of Life Sciences, South Kensington Campus, Imperial College London, London SW7 2AZ, UK; j.jimenez@imperial.ac.uk; 6Division of Cancer, Department of Surgery and Cancer, Imperial College London, Hammersmith Hospital Campus, London W12 0NN, UK; jonathan.krell@nhs.net; 7Department of Medical Oncology, VU University Medical Center, Cancer Center Amsterdam, 1081 HV Amsterdam, The Netherlands; e.giovannetti@amsterdamumc.nl; 8Fondazione Pisa per la Scienza, 56017 San Giuliano, Italy

**Keywords:** pancreatic ductal adenocarcinoma, microbiome, mycobiome, FMT, immunology, biomarkers, chemotherapy, PDAC, pancreatic cancer

## Abstract

**Simple Summary:**

This review summarizes the current literature related to the microbiome and pancreatic ductal adenocarcinoma (PDAC). The aim of this review is to explore the current role of the microbiome in the disease process, screening/diagnostics and to postulate the future role with regards to therapeutic strategies including chemotherapy, immunotherapy and surgery. We further explore the future of microbiome modulation (faecal microbiome transplants, bacterial consortiums, anti-microbials and probiotics), their applications and how we can improve the future of microbiome modulation in a bid to improve PDAC outcomes.

**Abstract:**

Pancreatic ductal adenocarcinoma (PDAC) is expected to become the second most common cause of cancer death in the USA by 2030, yet progress continues to lag behind that of other cancers, with only 9% of patients surviving beyond 5 years. Long-term survivorship of PDAC and improving survival has, until recently, escaped our understanding. One recent frontier in the cancer field is the microbiome. The microbiome collectively refers to the extensive community of bacteria and fungi that colonise us. It is estimated that there is one to ten prokaryotic cells for each human somatic cell, yet, the significance of this community in health and disease has, until recently, been overlooked. This review examines the role of the microbiome in PDAC and how it may alter survival outcomes. We evaluate the possibility of employing microbiomic signatures as biomarkers of PDAC. Ultimately this review analyses whether the microbiome may be amenable to targeting and consequently altering the natural history of PDAC.

## 1. Introduction

Pancreatic ductal adenocarcinoma (PDAC) is expected to become the 2nd most common cause of cancer death by 2030 in the USA. Progress in improving PDAC survival continues to lag behind that of other cancers. In the USA, 5-year survival was 2% of cases diagnosed in 1975–1977, progressing to 4% (1987–1989), 6% (2003–2009) and 9% in 2009–2015 [1,2]. This poor prognosis relates to the relatively late stage at which PDAC is first diagnosed, with most cases presenting with locally advanced or metastatic disease [3]. Currently in the UK, only 7% survive to 5 years, with just 25% managing to survive to 1-year [4]. Despite improvements in surgical techniques, perioperative care, chemotherapy, and radiotherapy, there has been little progress in improving survival outcomes. Nevertheless, a minority of patients do manage to survive to, and beyond, 5-years after surgical resection. The tumour genome, however, does not seem to explain the improved survival in this subset of patients [5]. This suggests an alternative explanation for the survival difference. A better understanding of this difference is fundamental to improving PDAC outcomes.

PDAC, and the wider cancer field, have been viewed through Hanahan and Weinberg’s hallmarks of cancer since their seminal work was published in 2000 [6]. In it, they postulated six hallmarks of cancer with two further emerging hallmarks and two enabling characteristics added in 2011 [7]. Crucially, a model of reciprocal interactions between tumour and the stroma have been marked as important areas of future research [7]. For decades, these hallmarks have been the principal avenues pursued by cancer research. How the hallmarks of cancer are influenced by the microbiome have been the focus of previous work for various cancers [8,9,10,11]. Finally in 2022, Hanahan published a further update to this work, recognising the microbiome as an enabling characteristic that facilitates the acquisition of cancer hallmarks [12].

The microbiome has been recognised as playing a pivotal role in influencing the immune system, interactions with cancer therapeutics and outcomes in PDAC and other cancers [10,13,14,15,16,17]. In this review, we provide an overview of recent studies that have added to our understanding of the microbiome, and the role it plays in the current and future management of PDAC. We summarise the relevance of anatomically specific microbiome profiles in PDAC, including the oral, duodenal, biliary, pancreatic, and faecal microbiomes. Additionally, we consider the potential role of bacterial extracellular vesicles (BEVs) and bile acids in PDAC. We address the influence of the microbiome with therapeutic strategies targeting PDAC, including chemotherapy, immunotherapy, and surgical intervention. Finally, we consider future applications of microbial therapies, such as antibiotics, probiotics and faecal microbiota transplants (FMT). Ultimately, we propose that there is the potential to harness the microbiome to advance therapeutics, diagnostics, and/or prognostic tests to improve survival outcomes in PDAC.

### Definitions

As the field of microbiome research evolves, our consensus on definitions have evolved, too. The microbiome was eloquently described by Whipps et al. in 1988 as a “*characteristic microbial community occupying a reasonably well-defined habitat which has distinct physico-chemical properties. The term thus does not only refer to the microorganisms involved but also encompasses their “theatre of activity*” [18]. The theatre of activity refers to the entire spectrum of molecules produced by microorganisms (living or non-living), encompassing structural elements, metabolites, and molecules produced by co-existent hosts, all of which are structured by surrounding environmental conditions. Recently, Berg et al. have added to this definition, proposing that the microbiome forms “*a dynamic and interactive micro-ecosystem prone to change in time and scale, is integrated in macro-ecosystems including eukaryotic hosts, and here crucial for their functioning and health*” [19].

The study of microbiota comprises only the living members present in a defined environment, including bacteria, archaea, fungi, algae, and small protists. It does not include phages, viruses, plasmids, prions, viroids and free DNA, which are non-living members of the microbiome [20]. Despite terms such as mycobiome gaining popularity in the PDAC related field [21], Berg and colleagues have highlighted that using such terms may not be correct, as they are included in the microbiome [19]. Alternative terms for the mycobiome have been proposed, such as fungal community. Indeed, mycobiome refers to the specific study of the fungal community; however, “biome” suggests the investigation of a specific environment and the interaction of its shared members. Whilst acknowledging this distinction, given the term’s prevalence in the PDAC literature, we will refer to the mycobiome as the fungal community in this article.

Bacterial taxonomy has facilitated our understanding of the bewildering diversity of the microbiota. In bacterial taxonomy, any given bacterium is collected into homogenous taxonomic groups, based on shared phenotypic or genotypic characteristics. These characteristics are assessed, and bacteria are subsequently grouped up the taxonomic ladder. All bacteria are organised in ascending levels as follows: species, genera, family, order, class, phylum, and domain; where species is the smallest unit, and domain the largest. Advances in technology have allowed us to gain efficient and comprehensive knowledge of the human microbiome. In fact, the human body harbours about one prokaryote for every human somatic cell, with a great and diverse variety in the over 1000 distinct species that colonise us [22,23,24]. This results in a wide array of genetic variability. We are gaining an extensive catalogue of genera and species that reside on and in us, with potential functional importance at both the individual microorganism and the community level [25]. A comprehensive explanation of the categorisation of bacteria is beyond the scope of this review.

## 2. Discussion

### 2.1. The Pancreatic Intra-Tumoural Microbiome and Its’ Relationship with Metabolism, Immune Response and Survival Outcomes

#### 2.1.1. The Bacterial Microbiome in PDAC

Recent research has uncovered the presence of clinically relevant populations of bacteria within the pancreas, biliary tree, and bowel in patients with PDAC [26,27,28,29,30,31]. There is growing evidence that: specific bacteria are present in PDAC [16,21,26,30,31,32,33,34,35,36]; the pancreatic intra-tumoural microbiome may have shared characteristics with the intestinal microbiome; and the PDAC microbiome is clearly distinct to that of normal pancreatic tissue [16,21,30,31,32,34]. Indeed, bacterial classes classically found in the duodenum, such as *Gammaproteobacteria*, are commonly found in human PDAC tissue samples [26,30,32,33,34,35]. There is likely a relationship between duodenum, gut, and pancreatic tumour microbiomes; however, the route(s) underlying colonisation of the pancreas remain debated [13,37]. A growing body of evidence suggests that one such route is translocation from the gut, specifically from the duodenum retrograde up the bile and pancreatic ducts, thereby, allowing colonisation of the pancreatic microenvironment [13,21,32,34,38].

Some authors have proposed gastrointestinal spread via the portal circulation and/or mesenteric lymph nodes; however, the mechanisms underlying this are yet to be fully elucidated [13]. Others have proposed systemic bacterial metabolites such as bile acids, fatty acids or polyamines derived from the gut microbiome may induce carcinogenesis or tumour progression at distant sites/organs [39]. Below, we discuss bacterial extracellular vesicles (BEVs), which warrant further investigation as a means for inter-kingdom communication between PDAC and the gastrointestinal microbiome. We believe that the colonisation of PDACs occurs through a variety of mechanisms, including, but not limited to, translocation via the pancreatic duct, systemic bacterial metabolite effect, the portal circulation and/or mesenteric lymph via messengers.

Irrespective of the underlying mechanisms, bacterial and fungal colonisation of the pancreas has been demonstrated to be associated with PDAC patient outcomes and may explain survival differences [16,21,34,36,40]. Riquelme and colleagues looked at the microbial metagenome of long-term survivors (LTS) of PDAC and their short-term survivor (STS) (survival greater than five-years versus death within five-years) [16]. In their study, tumour microbiome characteristics were similar to those previously described by Geller et al. [32]. They found a significantly higher alpha diversity (*p* < 0.05) in the PDAC microbiome of LTS compared to STS, which was validated in a separate cohort [16]. Alpha-diversity is a summary statistic of the distribution of taxonomic group abundance within a given community and/or the number of groups, ultimately showcasing heterogeneity in a given sample as a single number [41]. Interestingly, in the PDAC microbiome *Pseudoxanthomonas*, *Saccharopolsypora* and *Streptomyces* were shown to be significantly more abundant in the LTS cohort [16]. These three taxa demonstrated excellent discriminatory power in identifying survivorship (area under curve (AUC) receiver operator characteristic (ROC) 88.89% in the discovery cohort, and 86.67% in the validation cohort). The presence of these three taxa in the PDAC microbiome may be predictive of a more positive prognosis with a high degree of specificity, irrespective of the genomic make-up of the tumour. Adding *Bacillus clausii* to the three previously mentioned bacterial taxa brought the AUC values up to 97.51% and 99.17%, respectively [16]. Using tumour immunological profiling, Riquelme et al. went on to demonstrate a positive correlation between tumour microbiome biodiversity, the 4 taxa described and CD8^+^ T-cell density (measured via immunohistochemistry and multiplex immunofluorescence) in the tumours of LTS [16]. T-cell immunity is a recognised factor in long-term survivorship of PDAC, with LTS demonstrating high levels of CD8^+^ T-cell tumour infiltration, Th1-related gene expression, and M1 macrophage differentiation in previous studies [16,34,42,43].

Previous works have confirmed the molecular subtyping of PDAC and its association with prognosis [44,45,46,47]. The prospect of the tumour microbiome being associated with specific PDAC subtypes has recently been explored [36]. Guo et al. (2021) have used metagenomic sequencing to identify specific tumorigenic microbiome compositions associated with PDAC subtypes [36]. It was demonstrated that basal type tumours have a more diverse microbial community compared to other less aggressive phenotypes. However, it is becoming clear that whilst microbiome heterogeneity may play a role in tumour prognosis, the abundance of specific taxa within the microbiome is associated with positive or poor prognosis [16,36]. Basal-like tumours showed an increased abundance of *Acinetobacter*, *Spingopyxis* and *Pseudomonas genu.* This agrees with previous work that has shown that *Pseudomonas genu* may be associated with poorer outcomes [16]. Indeed, there may be scope to use the specific microbiome as a predictive tool for PDAC outcomes [16,36]. Their work also suggests that an immune infiltration of memory B cells, follicular helper T cells and activated mast cells were significantly higher in the basal subtype (an aggressive sub-type with poorer prognosis).

Pushalkar and colleagues examined the immune infiltrate of PDAC in KC mice following microbial ablation [34]. KC mice bear a mutant allele of K-Ras (LSL.G12D), with expression limited to pancreatic cells via a PDX-1-Cre. This gives rise to a mouse model with oncogenic K-Ras expressed predominantly in the pancreas, leading to tumours that progress from pre-invasive PanIN to invasive and metastatic PDAC at a low frequency, reflecting a sequence observed in de novo human PDAC [48,49]. In their study, ablation of the microbiome was achieved via an antibiotic cocktail consisting of Vancomycin, Neomycin, Metronidazole, Amphotericin and Ampicillin. This resulted in increased CD8^+^ T-cell infiltration, Th1 polarisation of CD4^+^ T-cells, and M1 macrophage differentiation in the tumours of microbially ablated mice, correlating with an enhanced anti-tumour immune response [34]. Conversely, the relative abundance of other intra-tumoural bacterial genera (mainly members of the class *Gammaproteobacteria*) correlates with metastatic disease, decreased immune infiltration and poor prognosis [33]. In combination, these results raise the possibility that the composition of the PDAC microbiome may alter immune infiltration and the milieu of immune cells that infiltrate the tumour. It also suggests that the microbiome can be modified, altering the immune infiltrate via the application of antibiotics, with the potential to change the natural history of PDAC.

Identifying the immunological processes that may induce this immune switch have been investigated via in silico models [50]. Neo-antigens are the resultant antigens from somatic mutations in cancer, leading to novel amino acid sequences and epitopes for major histocompatibility complex (MHC) expression. These antigens are novel and T-cells recognising these will not have been selected out during central tolerance [51,52]. Thus, T-cells recognising these escape detection and drive T-cell responses against cancer cells expressing neo-antigens on MHC [51]. In silico models have predicted LTS to possess both high neoantigen quality and quantity, with neither quantity nor quality alone correlating with LTS status. Additionally, patients with high neoantigen quality and quantity showed stronger infiltration and activation of CD8^+^ T-cells. Interestingly, the presence of neoantigens with homology to infectious disease-derived peptides identified LTS. These findings were corroborated in vivo, using samples of tumour infiltrating T-cells from LTS, which demonstrated cross reactivity with cancer and homologous non-cancer microbial antigens [50]. This raises the possibility of infectious agents (prior or concurrent) and their antigens, as a source for a T-cell reservoir capable of recognising and responding to cancer neo-antigens.

Neoantigen targeting vaccines have been shown to induce T-cell responses, in melanomas and other immunogenic tumours [53]. PDAC is not as immunogenic [54]. Using a neoantigen-based vaccine (PancVAX), in combination with the STING adjuvant, transient tumour suppression has been demonstrated in a mouse model. Addition of anti-PD-1 immunotherapy and an agonist antibody to OX40 gave rise to a durable anti-tumour response and a survival benefit in mice [52,53]. The depletion of CD8^+^ T-cells in these mice led to the complete loss of anti-tumour response, while only partial loss was observed with CD4^+^ T-cell depletion [52]. Crucially, the addition of OX40 agonist antibodies gave rise to a change in the CD4^+^ T-cell repertoire, shifting away from the T-reg phenotype (FoxP3^+^ CD4^+^) towards one expressing interferon gamma, promoting cytotoxic responses [52]. This raises prospects for PDAC patients in the future, with novel vaccine therapies potentially around the corner.

#### 2.1.2. The Fungal Community in PDAC

It has recently become apparent that the fungal community may also have a role in the PDAC TME. Using fluorescent in situ hybridisation staining with 28S rRNA probes, one group demonstrated a distinct composition of fungi in human and mouse PDAC models [21]. Another group has shown duodenal fluid aspirates in PDAC patients contain higher fungal DNA levels compared to healthy controls [55]. Interestingly, PDAC is associated with a three-thousand-fold increase in density of fungal species, compared to normal pancreata [21]. Alpha and beta diversity indices show distinct fungal composition in PDAC compared with that of the gut, with *Malassezia* spp. found in abundance in both KC mice and patients. Additionally, human PDAC bears a distinct fungal community to that found in healthy pancreata. Ablation of the fungal community in KC mice with amphotericin B produced smaller tumour weights, lower fractions of fibrosis and higher fractions of preserved acinar area. This suggests some fungal communities influence tumour growth. The pathways by which this is achieved have been interrogated [21].

MBL (mannose binding lectin) recognises fungal pathogens and binds fungal wall glycans in order to activate the lectin pathway of the complement cascade, triggering C3 convertase which cleaves C3, releasing subunit C3a, which interacts with C3a receptors found on tumour cells and, in doing so, promote tumour growth [56,57,58]. It has been shown that MBL expression is associated with reduced survival in human PDAC and that deletion of MBL protects against tumour growth in mice models [21]. It may be that translocation of fungi from gut mycobiome to the PDAC TME causes activation of MBL and subsequent complement cascade, which ultimately causes C3 associated tumour growth. This hypothesis is supported by evidence that recombinant C3a accelerates tumour growth in vivo. Moreover, the knockdown of the C3a receptor protects against tumour growth, but also abrogates the tumour suppressive effect of anti-fungal therapy, further strengthening the MBL-C3 axis hypothesis [21]. The fungal and bacterial constituents of the microbiome are in continual interaction [59]. We propose that the two should be considered in concert when targeted. Experimental data examining the possible synergistic effects of simultaneously ablating both compared with either strategy in isolation in animal models of PDAC are urgently required.

#### 2.1.3. The Effect of the PDAC Microbiome on Metabolism

There has been significant study of the microbiome in various disease states. Microbiomic differences have been demonstrated to predispose and contribute to a variety of disease states, including obesity, type-2 diabetes, cardio-metabolic disease, and non-alcoholic liver disease via systemic metabolic shifts [60]. Equally, metabolic dysfunction can induce changes in the microbiome, suggesting dynamic communication between host and microbiome [11,61]. Riquelme and colleagues used the Kyoto Encyclopaedia of Genes and Genomes’ (KEGG) pathway maps and modules to model the metabolic pathways present in the microbiomes of LTS and STS PDACs tumours [16]. Analysis showed differential clustering between LTS and STS, suggesting distinct pathways represented in the two groups. LTS microbiomes showed enrichment in the metabolism of amino acids, xenobiotics, lipids, terpenoids and polyketides, in addition to other cellular functions. The enrichment of xenobiotic biodegradation and lipid metabolism pathways correlated with better patient survival outcomes (hazard ratios 5.198 and 4.528 respectively). Meanwhile, STS demonstrated upregulation of protein synthesis and processing, genetic information processing, energetic and nucleotide metabolism, DNA replication, and repair pathways. These findings suggest a difference in the metabolic profile of the microbiome within the PDAC tumours of LTS and STS. Other work has shown PDAC Basal-type to be associated with increased levels of metabolism, energy production, conversion, replication, defense mechanisms and cell membrane/envelope biogenesis [36]. The microbial genu associated with Basal-type PDAC have been shown to positively correlate with some of these functions, such as DNA replication and Kras signalling and other pancreatic cancer related pathways. Furthermore, Basal-type PDAC has been shown to be associated with microbial abilities of metabolic activity, cell motility and antibiotic resistance [36]. Such a hostile microbial environment associated with aggressive phenotypes must be considered as we explore the role of microbiome-modulating therapeutics and their clinical application. The interaction between microbiome, its metabolic products (metabolome) and the changes these induce in the host are a field of active study.

We propose that the microbiome contributes to survival differences via the complex interaction of various pathways. Invasion of the pancreas by certain species may generate certain metabolites in the tumour, generating signals within the TME favouring a particular cancer phenotype through immune pathways. Metabolic inhibitors, such as the folate antagonist methotrexate, show effective inhibition of T-cell subpopulations and are clinically used explicitly for this effect [62]. The microbiome may produce factors that either inhibit CD8^+^ T-cells or induce a Th2/Treg rich immune phenotype, thereby generating a tumour-permissive environment. Equally, in LTS, the microbiome may provide certain neoantigens that allow for the molecular mimicry of tumour antigens, leading to immune activation, in a similar manner to certain forms of autoimmune disease, thus, resulting in cancer immune surveillance. Finally, there may be an aspect by which, during tumourigenesis, tumour promoting mutations occur in a distinct sequence, which creates a microenvironment favouring pancreatic colonisation by certain taxa, which are, in turn, noted to correlate with prognosis. These infect the tumour and establish a tumour permissive or regulating niche within the TME. Guo et al. (2021) have suggested that host genetics create an immune imbalance that facilitates the invasion of pathogenic microbiota that promote carcinogenesis [36].

We believe that a dynamic combination of the aforementioned processes likely occurs within the tumour-microbiome interface, with crosstalk impacting the immune response. We postulate that the microbiome in PDAC-phenotypes may contribute to certain cancer hallmarks, namely, deregulation of cellular energetics, avoiding immune destruction, and tumour promoting inflammation, resulting in shorter survival in these patients. Study of this is required in PDAC, with work providing some early evidence of this [16,34,36]. Similar models have been proposed in other cancers [10,11,12,34,63,64,65,66,67,68,69,70,71,72].

Indeed, in colorectal cancer, models of bacterial interactions in colorectal pathogenesis have been proposed. It has been postulated that “driver” microbes that initiate colorectal cancer development are followed by “passenger” microbes, which have a growth advantage in the environment established by the drivers [8,73,74]. This suggests that disease initiation changes the TME, consequently changing the microbial community, promoting tumour progression. Some have proposed that driver bacteria may function at the initiation stage, and passengers may be essential contributors to the promotion or progression stage [8,75]. We imagine, given the role that the microbiome plays in PDAC outcomes, that similar models would apply to PDAC and warrant investigation in this context, as well as in colorectal cancer. Indeed, Pushakar et al. [34] serially analysed the faecal bacterial profiles of KC mice compared with WT controls over time. In the early murine life, bacterial profiles were similar; however, diverging bacterial profiles eventually declared themselves, with growing divergence over time. Changes were also associated with disease progression, in keeping with previous work [21]. The temporal basis of these changes suggests a stepwise change that permits invasion by other species as new niches are generated in the gut by tumour associated changes. Further studies examining temporal changes in the pancreatic microbiome in animal models, as well as factors produced by potential ‘driver’ candidates, would be of significant interest to the PDAC field. It also raises the prospect of identifying pre-malignant states by screening for microbiomic changes.

#### 2.1.4. Effects of the Microbiome on the Immune Response in PDAC

Conventional cancer immunotherapy targets tumour-promoting inflammation and immune evasion by targeting immune checkpoint blockade, enhancing tumour-specific T-cell activity and anti-tumour T-cell education. Common immune checkpoint inhibitors include nivolumab, pembrolizumab, and pidilizumab (targeting programmed cell death protein-1, PD-1), atezolizumab (programmed cell death protein ligand-1, PD-L1), and ipilimumab and tremelimumab (cytotoxic T-lymphocyte–associated antigen 4, CTLA-4) [76]. These exhibit demonstrable efficacy in treating several solid organ malignancies and are widely employed [77]. Unfortunately, targeting T-cell immune checkpoint receptors or their cognate ligands have failed in PDAC clinical trials to date [78].

PDAC creates a highly heterogeneous, poorly characterised immunosuppressive TME (Figure 1). Additionally, the dense stroma found in PDAC inhibits the migration of cytotoxic T-cells towards tumour cells. These afford PDAC effective protection against chemotherapeutic agents and immunotherapy [53,78]. One key to unlocking the efficacy of immunotherapy in PDAC may be found in treating early-stage PDAC. Developing our understanding of the role of microbiota in PDAC initiation, progression, and immunosuppression is imperative for developing novel therapeutic strategies.

Microbial infections often lead to inflammation, a protective response, and the release of toxins [79,80]. Chronic inflammation, however, can lead to tumourigenesis through the activation of tumour-promoting cellular pathways. Indeed, patients with hereditary autoimmune pancreatitis are estimated to carry a lifetime risk of 40% of developing PDAC, while those with chronic pancreatitis pose a 13-fold higher risk of PDAC [81]. This is driven by chronic inflammation and proliferation of pancreatic stellate cells [82].

Additionally, microbes may play a crucial role in sustaining tumour cells. As part of the inflammatory response, antigen capture and presentation occurs via micropinocytosis, allowing the presentation of antigens on MHC for T-cell activation and responses to be initiated [83]. More recently, it has been shown the Wnt (Wingless/Integrated) signalling pathway, which is important for cell proliferation and differentiation during PDAC tumourigenesis, is also associated with micropinocytosis of microbes in cancer [84]. This is significant, as PDAC is recognised to grow in a nutrient poor environment; it has been shown to overcome this by actively scavenging extracellular proteins for growth via micropinocytosis [85]. The other important mechanism used by PDAC to overcome lack of nutrients is the adoption of an autophagic phenotype. Studies are required on the contribution (if any) of PDAC microbiomic products as an accessible source of nutrients to sustain PDAC growth.

Mechanistically, the microbiome in PDAC applies strong suppressive influences on the inflammatory TME. Activation of Toll-like receptors (TLRs) on myelo-monocytic cells expands MDSC and anti-inflammatory tumour-associated macrophages (TAM) populations [64]. As shown in Figure 2, these promote CD4^+^ differentiation of T-cells and supress cytotoxic CD8^+^ T cells. Vetizou et al. demonstrated the key role *Bacteroides* play in the immunostimulatory effects of CTLA-4 blockade, enhancing melanoma control as compared to germ-free or antibiotic treated mice [62].

Lipopolysaccharide (LPS) stimulation of TLRs links the microbiome to inflammation and tumourigenesis. LPS is a Gram-negative bacterial cell wall component and is recognised by the pattern recognition receptor TLR4 [86]. The interaction between LPS and TLR4 generates a cytokine chain, inducing IL-1β production and the infiltration of immunosuppressive lymphoid and myeloid cells, reducing the number of intratumoural cytotoxic CD8^+^ T cells. This activates the STAT3 (signal transducers and activators of transcription 3) pathway and activates mutated Kirsten rat sarcoma viral oncogene (KRAS), promoting PDAC progression [87].

Modulating the microbiome to enhance the efficacy of immunotherapy has been demonstrated in animal studies. Pushalkar et al. reported an association between the microbiome and immunotherapy in a PDAC mouse model, showing synergy between antibacterial treatment and anti-PD-1 therapy on tumour size [34]. Interestingly, PD-1 expression was upregulated in T-lymphocytes in the antibiotic treated mice. Given the T-cell suppressive role of PD-1 expression, the addition of PD-1 blockade to the antibiotic treated group was studied. Antibiotics in combination with PD-1 blockade demonstrated improved anti-tumour response. Moreover, PD-1 blockade with antibiotics produced smaller tumour sizes as compared to antibiotics alone, whereas PD-1 blockade alone did not impact tumour growth [34]. This suggests a degree of synergistic anti-tumoral activity between microbial ablation and PD-1 blockade. Whilst antimicrobials have sparked interest in improving the efficacy of immunotherapy in PDAC, Lee et al. advise that the use of broad-spectrum antibiotics, proton pump inhibitors or over-consumption of animal meat or lack of a plant-based food consumption may hold detrimental implications on the gut microbiome and subsequent efficacy of immunotherapy [88]. Whilst their recommendations are based on observation and short-term clinical data, the role of microbiologists and dieticians with expertise in the gut microbiome should not be overlooked when considering initiating immunotherapy.

Clark et al. noted that immunosuppression in PDAC is associated with the presence of tumour-promoting immune cells rather than inflammatory cells [89], whereas, patients with a higher mortality in PDAC exhibited a low T-cell infiltration with a secretion of CXCL12 chemokine and CAF expression of fibroblast activation protein (FAP) [88]. CXCL12 has been found to limit T-cell infiltration [89]. Microbial signals can activate pancreatic stellate cells that release IL-1β from PDAC cells [90]. Similarly, focal adhesion kinase 1 (FAK1) proliferates collagen I deposition and both actions suppress the immune response. This, in turn, reduces tumour infiltration by CD8^+^ cytotoxic T-cells and correlates with a poor survival in PDAC [91]. B cells are important immunomodulators in PDAC. Within the TME, B cells upregulate immunosuppressive cytokines (IL-10, IL-18 and IL-35) and immune checkpoint ligands (particularly PD-L1) [92]. This inhibition of T-cell mediated tumour immunity promotes cancer immune evasion and, thus, leads to tumour progression [93]. However, it is also capable of producing oncogenic KRAS-specific IgG, which can contribute to an anti-tumour antibody treatment response.

Dendritic cells (DC) are antigen-presenting cells (APCs) that can generate tumour-protective T-cells within the TME. Sivan et al. investigated melanoma growth in mice [62] and showed that oral administration of *Bifidobacterium* improved tumour control to the same degree as PD-L1-specific antibody therapy, and that combination therapy nearly eradicated tumour outgrowth [94]. This effect was mediated by augmentation of DC function, enhancing CD8 T-cell priming. However, in PDAC DC, numbers are likely too small to replicate such a response in adaptive immunity within the PDAC TME. Hedge et al. identified that DC dysfunction in PDAC, both systemically and in tumour-draining lymph nodes, contributes to impaired T-cell priming [95]. CD40 is a costimulatory receptor that is expressed by myeloid cells in the TME. CD40 agonists potentiate the immunogenicity of DCs, convert TAMs to a tumoricidal phenotype and reverse tumour-associated fibrosis [96]. Long et al. recently illustrated enhanced chemotherapy efficacy in PDAC with the use of CD40 agonists [97]. Another tactic is to combine T-cell targeted treatments with agents that can reverse immunosuppressive myeloid cells produced by tumour cells via stopping receptor–ligand interactions [98].

### 2.2. Influence of the PDAC Microbiome on Chemotherapy

It has been suggested that the presence of gut or intra-tumoural microbiota may modulate the host response to chemotherapy treatment. It may be that we can target specific microbiomic constituents to improve chemotherapeutic efficiency or use microbial biomarkers to predict treatment response.

For example, one of the first line chemotherapeutic agents used in PDAC, Gemcitabine (GEM; 2′,2′-difluorodeoxycytidine), has been shown to be affected by the enzymes pyrimidine nucleoside phosphorylase and cytidine deaminase [99]. In 2017, Geller et al. showed that *Gammaproteobacteria* can metabolise GEM into its inactive form (2′,2′-difluorodeoxyuridine) via expression of an isoform of cytidine deaminase (CDD). A mouse colorectal carcinoma xenograft model treated with GEM and antibiotics (ciprofloxacin) targeting *Gammaproteobacteria* displayed a marked anti-tumour response compared to control-treated mice, which instead developed tumour progression [32]. Human tissue samples of PDAC have identified *Gammaproteobacteria* as one of the most abundant classes in the PDAC microbiome [26,32]. Furthermore, bacterial cultures from PDAC tissue subsequently inoculated into human colorectal cancer cell lines rendered the cells fully resistant to GEM therapy. It has also been shown that an abundance of *Proteobacteria* correlates with metastatic disease, decreased immune infiltration and poor prognosis in PDAC models [33]. Indeed, it has been hypothesised that pancreatic intra-tumoural *Gammaproteobacteria* plays a key role in GEM resistance and an association with disease progression in PDAC [14].

It has been shown that the majority of bacteria (93%) cultured from patients who required instrumentation of their pancreatic duct were able to confer resistance to gemcitabine on tested cell lines [32]. It would be interesting to investigate whether instrumentation of the biliary tree is associated with translocation of gut bacteria such as *Gammaproteobacteria*, and whether this poses any implication on subsequent chemoresistance and outcomes in human studies. Studies investigating chemoresistance and microbiome composition in PDAC tissues between patients who underwent biliary stenting prior to pancreaticoduodenectomy (PD) and those who underwent “fast track” surgery may provide insights to help answer these questions.

Interestingly, the treatment of orthotopic PDAC mice treated with GEM and amphotericin B significantly augmented the effects of GEM with regards to tumour weight, as compared to control mice treated solely with GEM, amphotericin B or neither. This suggests an association between fungi and GEM resistance [21]; thus, comparative studies investigating the role of antifungal and antibacterial treatments in combination with GEM should be considered.

Whilst the relationship between GEM and the microbiome is of great interest, recent clinical trials in PDAC have demonstrated FOLFIRINOX (5-FU, Leucovorin, irinotecan and oxaliplatin combination) to offer improved survival outcomes compared to GEM [100,101]. Garcia Gonzalez and colleagues (2017) carried out work using the C. elegans nematode (a model for host-microbial interactions) in order to study how bacteria affect the response of chemotherapeutic agents such as fluoropyrimidines (e.g., FUDR & 5-FU) and topoisomerase I inhibitors (e.g., camptothecin) [102]. They have shown that bacteria can modulate drug responses and that different agents within the same class (i.e., FUDR & 5-FU) are differentially affected by bacteria. For example, while *E. coli* increases FUDR efficacy, it holds no role in increasing the efficacy of 5-FU. Another study in mice has demonstrated that antibiotic alteration of the gut microbiome is associated with decreased 5-FU efficacy in the treatment of colorectal cancer xenografts, and that genes involved in amino-acid metabolism, replication repair translation and nucleotide metabolism were found to cause reduced expression in the antibiotic treated group [103]. Furthermore, in mice treated with antibiotics that conferred a lower 5-FU anti-tumour response, the relative abundance of bacteria belonging to the *Proteobacteria* phylum was dramatically higher compared to controls [103]. Zhang et al. showed an association between *Fusobacterium nucleatum* and decreased 5-FU efficacy in colorectal cancer models, the proposed mechanism being upregulation of anti-apoptotic signalling and inhibition of 5-FU-induced cell apoptosis, by inducing BIRC3 expression via the TLR4/NF-kB pathway [104]. Furthermore, both oxaliplatin and 5-FU chemoresistance in colorectal cancer has been linked to *Fusobacterium nucleatum* [105]. Yu and colleagues have also shown that, in colorectal cancer models, *Fusobacterium nucleatum* may activate TLR4 and MyD88 immune signalling pathways, as well as downregulation of specific microRNAs in order to activate the autophagy pathway and reduce cell apoptosis induced by oxaliplatin and 5-FU. This is an area of interest that warrants further research, as *Fusobacterium nucleatum* is one of the most enriched species in the human PDAC microbiome [35]. In contrast, the culture supernatant of *lactobacillus plantarum* was shown to have a positive effect, dampening 5-FU chemoresistance in colorectal cancer through the inhibition of cancer stem-like cell formation [106]. Modulating the microbiome to improve the efficacy of oxaliplatin has also been demonstrated in mice bearing lymphoma xenografts [107]. It has been shown that certain bacteria confer tumour resistance to oxaliplatin; however, the mechanism is unknown [32].

Modulating the microbiome may also have a role in minimising adverse effects associated with chemotherapy, which may indirectly affect the efficacy of treatment. For example, mucositis is a severe and significant complication of chemotherapy, which can hold implications on patient quality of life, length of hospitalisation following chemotherapy and risk of death [108]. It can also lead to cessation of chemotherapy [108]. Irinotecan contains an active antitumour metabolite called SN-38 that is deactivated and expelled through bile into the gut in the form of SN-38G. β-glucuronidase enzymes produced by bacteria within the gut lumen can reactivate SN-38G into its enterotoxic form, leading to mucositis [37]. Recently, ciprofloxacin has been shown to suppress bacterial β-glucuronidase activity [109]. This warrants further investigation with regards to clinical application. We must, however, remain judicious in the use of antibiotics that indiscriminately deplete the gut microbiome [14], and be cognisant vis-a-vis antibiotic resistance and other infections (e.g., *Clostridium difficile* diarrhoea) in patients undergoing chemotherapy.

As we unearth the bacteria and fungi involved with drug disposition, action, and toxicity, we may be able to inactivate/activate them in order to potentiate chemotherapy, alter toxicity or alter the progression of disease. However, this research is very much in its early stages. Choy et al. have expressed caution that experimental models exploring the link between microbiome and chemoresistance may not be generalisable [110]. It is suggested that, in order to target the microbiome in future chemotherapeutic approaches, we must first understand the influence of host, environmental and local tumour tissue factors on microbiome composition and function [14,64,110]. There is a need to develop more robust evidence, specifically in PDAC, on the clinical application of chemotherapeutics co-administered with antibiotics and/or antifungals, assessing their impact on cancer outcomes. As we begin to understand the host, environmental and tumour specific influences on the TME, we may be able to explore individualised pharmaco-microbiomics and treatment strategies based on biomarkers and patient demographics. In Figure 3, we portray current and future strategic avenues for the microbiome in PDAC chemotherapeutic strategies, as well as barriers to success.

### 2.3. Faecal Microbiota Transplant in PDAC

#### 2.3.1. Evidence for Faecal Microbial Transplant in PDAC

If the constituents of the gut microbiome can affect the colonisation of the PDAC tissue [21,32,34,38], we must ponder altering gut microbiome in order to modify the tumour microbiome. It has been shown that microbiota from human faecal microbial transplants (FMT) administered to murine PDAC models can be detected in the PDAC microbiome post transplantation, albeit in small quantities of less than 5% [16]. Nonetheless this suggests that, similar to the host gut microbiome, transplanted microbiota can translocate into the PDAC tumour or cross talk and alter the TME. In fact, FMT yielded from PDAC LTS transplanted into murine PDAC models has been shown to significantly reduce tumour growth [16]. This positive effect of FMT with LTS material was then lost with antibiotics. Furthermore, after FMT, there were clear differences in the tumour microbiome between the antibiotic treated and antibiotic untreated mice. These differences also translated into changes in tumour immune cell infiltrates, with the non-antibiotic treated LTS FMT group showing a CD8^+^ T-cell rich environment, whilst the STS and healthy control FMT groups demonstrated increased numbers of CD4^+^FOXP3^+^ T-regs and myeloid derived suppressor cells, both of which are associated with dampening of immune responses, thereby permitting tumour growth [16]. Interestingly, when comparing the tumour microbiomes in mice receiving FMT from either LTS or STS and healthy controls, they found the gut and tumour microbial beta-diversity in all three mice groups differentially clustered. Moreover, mice with STS FMT possessed larger tumours than their healthy control FMT counterparts, further suggesting that the PDAC-associated gut/tumour microenvironment may hold a tumour-promoting effect.

Additionally, there is evidence that FMT preparations derived from KPC mice accelerates PDAC growth when transplanted into germ free or antibiotic treated KC mice [34]. KPC mice bear mutated LSL-KrasG12D and LSL-Trp53R172H alleles, driven by a Pdx1Cre, meaning pancreatic cells bear heterozygous P53 and oncogenic KRAS [49]. The resultant mouse model presents with very aggressive PDAC [49]. Furthermore, *Malassezia* spp. has been used to repopulate the gut of amphotericin B treated orthotopic PDAC bearing WT mice and shown fungal species can accelerate tumour growth [21]. Results were not reproducible for other fungal genera such as *Candida*, *Saccharomyces* and *Aspergillus*. It is logical to consider that FMT outcomes may be related to the compositional and functional differences of the various microbial communities in the stool that may alter the TME and, in doing so, determine either a negative or positive effect on tumour inflammation, metabolic pathways and disease progression.

However, FMT in PDAC may not be a ‘one size fits all’ strategy; host-specific effects must be considered. For example, it has been shown that the human PDAC microbiome differs depending on gender and that specific pancreatic tumour microbes may cause contrasting effects on oncogenic and immune pathways depending on gender [33]. Whilst the underlying reason for this remains largely unknown, it has been demonstrated that sex hormones influence the murine gut microbiota [111]. We also know that the gut microbiome and many other host characteristics are heavily intertwined and that there may be both host and environment specific factors impacting gut microbiome composition and functionality [10,21,36,112,113,114]. Additionally, recent evidence indicates that the success of FMT in initiating lasting compositional changes on the host’s gut microbiome is dependent on both donor and recipient microbiome compatibility [115]. Recent work has highlighted that differences in the host microbiome could be responsible for the variability in response to FMT for other diseases [116,117,118,119]. It has also been shown that the bacterial profiles of PDAC tissues vary substantially between individuals with regards to the mean relative abundance of certain taxa [26]. If there is a relationship between gut and PDAC microbiome, it is reasonable to consider that the tumour microbiome and tumour response to FMT associated changes in gut or tumour microbiota poses host specific effects, too. Thus, we propose that certain hosts may be more susceptible than others to FMT depending on their gut and tumour microbiome profile.

We must continue to investigate the trends in the specific microbial constituents of FMT, gut and tumour samples, determining which promote an antitumor or protumour effect, as well as the specific biochemical and cellular mechanisms by which they do so. Through this we may be able to stratify donors and hosts by their bacterial profiles, establish a microbial consortium with pre-defined microbial parameters and target underlying mechanisms. If FMT is a proven therapeutic avenue for PDAC, investigating these trends would be of great value in identifying predictors of treatment success, whether that be an assay of host tumour, bile, or stool microbiota or microbiomic markers, in order to further profile both donor and recipients for matching. We imagine a future where we possess such a deep understanding of the host and recipient microbiota that we can exploit the host microbiome with therapies that are so personalised that they may exert opposing effects from one patient to the next. However, we must continue to explore a separate vein of study into the shared and consistent characteristics of the PDAC tumour microbiome between individuals, which may allow for the widespread use of strategies. Either way, with the microbiome increasingly recognised as a source of discrepancy in PDAC survival, the further investigation of the interplay between the host, tumour and microbiomic factors deserves urgent study [89,120].

#### 2.3.2. Future Role of FMT in PDAC

##### Optimizing FMT in Order to Improve Outcomes

We know that higher alpha diversity in the stool of long term PDAC survivors transplanted into PDAC-bearing mice is associated with tumour regression [16]. Interestingly a recent meta-analysis investigating FMT in inflammatory bowel disease has shown that the pooling of stool from multiple donors increases microbial diversity and is associated with enhanced remission rates [117]. We must investigate whether donor pooling to increase diversity correlates with improved outcomes in PDAC. Furthermore, short term specific engraftment of the pancreatic tumour with oral delivery of bacteria/faecal microbiome has been demonstrated in mice models [16,34]. However, we have not proven long term colonisation of the pancreatic tumour nor long term tumour response in murine models nor a role for FMT in PDAC human subjects. One study in metabolic syndrome has shown that the long term (3 months) engraftment of donor bacterial strains with FMT can co-exist with recipient bacterial strains [115]. We must investigate the temporal changes of the pancreatic tumour microbiome composition shifts over time following transplantation of FMT and whether one course of FMT exerts long lasting effects on the PDAC tumour. A recent meta-analysis investigating the role of FMT in recurrent *Clostridioides difficile* infection found that the effect of FMT on clinical outcomes significantly increases with repeated administrations of FMT [121]. Whilst FMT in PDAC is in its infancy, to date there are no comparative animal studies in FMT for PDAC exploring the impact of donor pooling, dosing, or regimen intensity.

Furthermore, only a minority (5%) of FMT donor bacteria are detected in the PDAC tumour microbiome post transplantation in murine models [16]. Whilst this was enough to evoke a tumour response, to guide future practice related to FMT, future work should investigate how tumour microbiota populations with variable proportions of donor FMT correlate with outcomes. Another question to answer is: what is the minimum quantity of specific microbes or composition of microbiota that are required to evoke a positive tumour response? Through this understanding, we may be able to provide the most appropriate FMT regimen and composition, whilst minimising any adverse effects associated with its administration.

##### Routes of FMT Administration

In addition to the hypothesis that gut bacteria translocate into or remotely affect the tumour microbiome, it has also been suggested that the lower GI tract and mesenteric nodes may serve as sites for bacterial seeding to the pancreas [13]. Of great interest is FMT via the colonic route, and whether this holds implications on the PDAC microbiome. If so, this may provide an alternative route of administration that can be employed depending on patient needs. We also know that instrumentation of the pancreatic duct may affect the tumour microbiome [26,32,38,122]. From this, we hypothesize that direct delivery of microbiota with proven anti-tumour affects through endoscopic retrograde cholangiopancreatography (ERCP) may play a role in future modulation of the PDAC microbiome and may improve on the fraction of transplanted microbiota present in the TME compared to other routes of administration. Modulation of the microbiome directly via the biliary tract, whether that be through the delivery of microbiota transplants or local antibiotics, is likely to limit undesired off-target effects.

##### FMT Donor Selection

The faecal microbiome make-up may change with the course of PDAC tumourigenesis [21,34]. If FMT from PDAC donors is the therapeutic avenue we pursue as a novel treatment option or research focus, the evidence that the faecal bacterial profile evolves with the disease process draws into question at what stage in the disease should PDAC FMT patient donors provide a stool donation and how does FMT yielded from patients at different linear stages in disease correlate with outcomes in FMT transplanted patients? Can we reliably repeatedly use the stool of one donor consistently over time or must we carry out an assessment of donor stool composition each time a donation is given? It might be that we can administer a defined bacterial isolate or a specific microbial composition with the ideal characteristics that is not necessarily derived from a donor but rather cultivated in a laboratory setting, in order to circumvent such a hurdle. Additionally, as our understanding of the temporal relationship of faecal/gut bacterial profiles and PDAC disease progression evolves, we may be able to use this information to develop a screening tool for PDAC. Work to date on stool biomarkers is discussed further below.

From a clinical perspective, donor selection and screening would be key to the implementation of FMT in PDAC. In one study, two immunosuppressed patients developed bacteraemia with an antibiotic resistant strain of E coli (ESBL) following the administration of FMT capsules derived from a single donor, with one associated mortality [123]. Furthermore, there is a concern that SARS-CoV-2 may be transmitted by FMT; additional safety precautions should account for this in future studies and clinical practice [124]. We are still a long way away from clinical implementation in PDAC; however, the safety profile of FMT should be carefully considered in future clinical trials. In addition to vigorous screening of donors, we must be prudent in FMT recipient selection, paying due regard to co-morbidity, immunosuppression and, as discussed above, the host microbiome.

##### Role of Antibiotics in FMT

Another clinical consideration is the impact of antibiotics in patients being treated with FMT. It does seem counterintuitive to use antibiotics in patients with FMT; however, it is well known that cancer patients are prone to infection that may require treatment. In PDAC murine models treated with FMT from long term human survivors with no evidence of disease, post FMT short-term antimicrobial therapy results in larger tumours than FMT-treated models without antibiotics [16]. Indeed, the reduction in the bacterial tumour environment may weaken the impact of FMT. Subgroup analysis of those who do and do not receive antibiotics in human trials investigating FMT in PDAC would provide valuable insight. Pre-treatment of patients with antibiotics prior to FMT may enhance FMT efficacy in other diseases; it has been postulated, though not proven, that antibiotics eliminate competition from host microorganisms [125]. Interestingly, Pushalkar et al. [34] showed that bacterial ablation in recipient mice improved colonisation by the donor stool microbiome. Antibiotics may serve to eradicate possible competition between host and donor microbiota.

##### Future of FMT in PDAC Conclusion

The research available to date is in its infancy and there is a paucity of data and knowledge on the role of FMT in PDAC. We must be cautious in the conclusions we draw from the limited evidence currently available. Preliminary studies are promising; we must begin considering the clinical application of FMT in PDAC. Robust large-scale studies will allow for more subtle relationships to be identified, as the relationship between gut microbiome, tumour microbiome and FMT is intricate, and modulation may have effects on a wider scale throughout the body. There is also great variability in the methodology underpinning the administration of FMT in the few studies on PDAC, and they are not entirely representative of clinical practice. Furthermore, studies performed in mice may not be relevant to human physiology, as only 4% of the bacterial metagenome is shared between mice and humans [126]. We must remember that much of the gut microbiota is affected by environmental factors, including murine housing conditions and genetic background. Further research is, of course, warranted to consolidate the evidence base of FMT in PDAC. Figure 4 below outlines the barriers to future implementation of FMT that future trials must consider, as well as the proposed mechanisms by which outcomes are affected by FMT.

### 2.4. Role of Antimicrobials in PDAC

The role of antibiotics in disrupting dysbiosis in PDAC merits further investigation, as a growing body of preclinical evidence suggests a role for antimicrobials in PDAC. Thomas et al. assessed the pathological grades of pancreata harvested from genetically engineered PDAC mouse models (KrasG12D/+, PTENlox/+ and Pdx1-Cre) treated with either a control or a cocktail of broad-spectrum antibiotics eradicating the intestinal and pancreatic microbiota. Statistically significant differences between groups were identified, with both a greater number of malignant pancreatic lobules and a higher degree of poorly differentiated PDAC identified in the control-treated group [40].

Furthermore, antifungals may be protective against oncogenic progression in both progressive PDAC murine models and aggressive orthotopic models [21]. This evidence is strengthened by Sethi et al. who showed that a combination of antibiotics and antifungals may deplete gut microbiota, restrain tumour growth in murine PDAC subcutaneous implants and limit the metastatic burden of PDAC in mice [127]. They demonstrated that microbial depletion using a combination of antibiotics and antifungals impacts the adaptive immune tumour response, increasing anti-tumour IFN-Y-producing cytotoxic T-cells and inhibiting the number of interleukin IL-17A and IL-10 producing pro-tumorigenic T cells. Pushalkar et al. investigated the possibility that antibiotics could impact the TME in a mouse model. Examining the immune infiltrate of PDAC from KC mice after microbial ablation with an antibiotic cocktail revealed increased CD8^+^ T cell infiltration, Th1 polarisation of CD4^+^ T cells, and M1 macrophage differentiation, compared to non-ablated mice, demonstrating an enhanced anti-tumour immune response.

A retrospective analysis of 148 patients with PDAC found that macrolide consumption may be associated with increased overall survival and progression free survival [128]. However, antibiotic use to eradicate microbiomes may also exert negative effects in both PDAC and other diseases [129,130,131]. Furthermore, several preclinical studies have reported negative effects from combined antibiotic and immunotherapy use [62,107,132,133]. We know that cancer types possess specific signature microbiome compositions, and that any given organism can exert differing effects on different cancers [34,35,94]. However, there is an imperative to understand the mechanism by which antibiotics mediate changes in the gut microbiome, the interplay between specific organisms (at the same, as well as different, anatomical sites) and malignancy, and the implications these pose on outcomes in various patient groups. Such understanding will ensure that we are targeting the tumour-promoting and sustaining aspects of the microbiome, without stimulating alternative pro-tumour pathways on both a local and systemic level. Although pre-clinical research shows promising work regarding PDAC microbiome modulation and the application of antibiotic therapy, it may be too overly simplistic to eradicate the PDAC or gut microbiome with broad spectrum antibiotics in the aim to improve prognosis. Doing that may end up causing off-site effects in other tissues, resulting in collateral damage. Other risks of antibiotic use include causing a rise in multidrug resistant bacteria and magnifying the toxic effects associated with long term antimicrobial use.

Whilst evidence suggests that antibiotics may play a beneficial role in PDAC treatment, that exact role and how it translates into clinical practice remains to be determined. As our understanding of the specific effects of microbiomic constituents grows, we may be able to modulate the microbiota with precision, either through systemic therapeutics or local delivery devices, circumventing offsite side-effects. For example, recent pre-clinical work in mice has alluded to the possibility of delivering antibiotics into tumours through the implantation of microdevices capable of releasing microdoses of therapeutics [32]. As we accumulate data on the microbiota and its relationship with many disease states, we could see a future wherein a patient is screened for a multitude of biomarkers; screening for and identifying patients who may harbour risk-associated or disease-defining microbiomic markers. This would allow for early disease identification and prevention. In doing so, we may also be able to plan therapies accordingly, targeting specific microbial constituents in specific sites, minimising collateral damage.

### 2.5. Modulation in the Perioperative Period

Surgical intervention for PDAC carries a notoriously high morbidity rate. We may be able to identify microbiota associated with poor outcomes in order to mitigate this risk. The advantages of this are twofold. Firstly, we may be able to prevent complications, although there are many variables to consider with regards to the aetiology of operative complications beyond the microbiome. Secondly, it may be possible to identify patients at risk of complications in order to strategize their pre- and post-operative care appropriately. This may provide huge benefits in patient care and cost containment.

The possibility of the gut microbiome predicting outcomes has been documented in colorectal surgery; however, whether the presence of these microbes is causative or associative remains unsolved. Van Praagh et al. isolated bacterial DNA from the site of colorectal anastomoses and demonstrated that anastomotic leak is strongly associated with low microbial diversity, as well as a high abundance of *Bacteroidaceae* and *Lachnospiraceae* [134]. Furthermore, samples with a microbial composition consisting of 60% or more of these two families predicted anastomotic leak in this study [134]. Shogan et al. have showed that *Enterococcus faecalis* may contribute to anastomotic leak in rat models, proposing this to be mediated by the degradation of collagen and activation of tissue matrix metalloproteinase (MMP9) in host intestinal tissue. This group also showed that topical antibiotics targeting *Enterococcus faecalis* directly applied to rat intestinal tissue prevented anastomotic leak [135].

In pancreatic surgery, a recent prospective pilot study found associations between the relative abundance of specific organisms in the stool before and after surgery and postoperative complications including pulmonary embolism, infection, fistula formation and delayed gastric emptying [136]. Whilst this was a relatively small study, it suggests that the microbiome can predict the likelihood of operative complications in some patients. Additionally, the presence of *Enterococcus* in drain samples has been associated with leakage in pancreaticoduodenectomy patients [137]. Another study demonstrated an association between post-operative pancreatic fistula formation (POPF) and an abundance of the commensal anaerobe *Bifidobacterium* within pancreatic fluid, as well an abundance of *Klebsiella* in stool [27]. This group speculates that POPF involves microbiota induced generation of cross-reactive antibodies that contribute to collagen degradation, in a similar vein to previous work described above [135]. It has previously been shown that *Klebsiella* generates cross-reactive antibodies against HLA antigens and collagen molecules in autoimmune disease [138]. Further work is needed to validate these preliminary results.

Decreased levels of *Klebsiella* in faecal samples post pancreatoduodenectomy may be associated with death at 1 year post pancreatoduodenectomy [27]. This contrasts with the association between an abundance of stool *Klebsiella* and POPF [27]. Indeed, any given organism, in either high or low abundance, may exert deleterious effects on various outcomes. This, again, highlights the intricacy of the microbiome and the possible risk of collateral damage when altered, which must be carefully considered when modulating the microbiome. Ideally, we would only induce favourable effects. In reality, we may find early clinical application of microbiomic modulation associated with compromise and minor adverse effects. In the example of *Klebsiella*, given the adverse effects from increasing and decreasing *Klebsiella* stool abundance, instead of targeting *Klebsiella* itself, we may find better results in targeting the cross-reactive antibodies that *Klebsiella* induces, which cause collagen degradation. Thus, we may be able to promote the positive effect *Klebsiella* promotes and mitigate the unintended consequence of POPF.

To date, studies have mostly focused on identifying bacteria associated with operative complications. Of equal importance, however, is the identification of bacteria that promote healing and operative success. This would further the application and relevance of therapies like FMT and probiotics and help guide antibiotic choices in the perioperative surgical patient. Blind eradication or promotion of the microbiome would be most unwise. We keenly await the results of the PANDEMIC study (NCT04274972), a prospective, observational, cohort study currently underway, assessing the qualitative and quantitative analysis of the pancreatic microbiome in human patients with PDAC undergoing pancreaticoduodenectomy and intraoperative lesion sampling. Outcomes to be assessed include correlations between bile, oral and rectal microbiome samples and the correlation between the pancreatic microbiome and development of post-operative complications [139].

### 2.6. Biomarkers and Diagnostics

The work into microbial biomarkers in PDAC is very much in its infancy; however, studies to date are encouraging and hint towards various possible applications in PDAC, in a bid to improve outcomes (Figure 5). We will discuss below the association between site specific microbiota and PDAC, as revealed in current studies.

#### 2.6.1. Oral Microbiota

The oral cavity contains an abundance of different types of bacteria, viruses and fungi [140]. When composition of the oral microbiome changes, it can become pathogenic and be associated with diseases like PDAC and others [39,141]. A recent meta-analysis of eight studies suggested a significant link between periodontal disease and risk of PDAC [142]. They have suggested that this link may be associated with the oral microbiome. Independent risk factors for PDAC included poor oral health, periodontal disease and tooth loss [143,144,145]. Some suggest that this may be via translocation from mouth to gut to pancreatic tissue [39].

Farrell et al. analysed the oral microbiome of patients with resectable PDAC and matched controls to assess for markers of PDAC [146]. Two of the six bacterial candidates (*Neisseria elongata* and *Streptococcus mitis*) were validated, with lower levels of these two species in patients with PDAC compared to controls (*n* = 56). These two salivary bacterial markers in combination were able to differentiate PDAC patients from healthy controls (ROC 0.90, CI 0.78 to 0.96, *p* < 0.0001) with 96.4% sensitivity but 82.1% specificity. The inverse relationship between *S. mitis* and PDAC has been shown in another study [147].

Michaud and colleagues conducted a study measuring antibodies to oral bacteria in pre-diagnosis blood samples for PDAC patients and matched controls. Results suggested that patients with high levels of antibodies against *Porphyromonas gingivalis* from chronic periodontitis carried a two-fold higher risk of PDAC [147]. *Fusobacterium*, conversely, was shown to be associated with decreased PDAC risk. Equally, antibodies against specific commensal organisms may reduce the risk of PDAC [147], whilst *Aggregatibacter actinomycetemcomitans* may be linked to increased risk of PDAC [148]. Most of the studies described are case control studies; therefore, a direct causative role for such bacteria in PDAC has not been demonstrated. Nevertheless, it has been known that pathogenic oral bacteria can induce oncogenic hits. *P. gingivalis*, *Treponema denticola*, and *Tannerella forsythia* are the chief pathogens that cause periodontitis. Such pathogens secrete peptidyl-arginine deiminase (PAD) enzyme; its degradation results in p53 and K-Ras point mutations [149]. Other work has explored the role of microbial biomarkers for PDAC associated with saliva or the tongue coating [150,151].

As discussed above, bacteria introduced orally can influence the pancreas TME [21,34,39]. There is likely a complex and dynamic relationship between environmental risk factors for PDAC, such as smoking, obesity and diabetes and the microbiome in various sites, including that associated with PDAC. These factors may aggravate disease initiation and progression through specific changes in microbiota compositions in the oral cavity, as well as other anatomical sites.

#### 2.6.2. Duodenal Microbiota

Investigating the source of bacterial translocation, Fritz et al. found the small bowel to be a major source of enteral bacteria in infected pancreatic necrosis in mice [152]. *Helicobacter pylori*, a familiar bacterium that colonizes the human stomach, has now come under scrutiny as a cause of PDAC. One hypothesis proposes that *H.pylori* causes a surge in hyperchlorhydria that increases the release of secretins, in turn promoting pancreatic hyperplasia [153]. Another thought is that hyperchlorhydria leads to atrophic gastritis, which leads to bacterial overgrowth and increasing carcinogenic levels of N-nitrosamines [154]. Mei et al. detected higher duodenal levels of *H. pylori* in PDAC patients compared to controls [28].

Aksintala et al. describe how gastrointestinal microbial dysbiosis contributes to the pathogenesis of pancreatic disease by chronic proinflammatory changes in the pancreas [155]. They highlight that gut microbes are integral in the modulation of gut architecture and for maintaining mucosal integrity, and that dysbiosis leads to the loss of gut mucosal barrier integrity. In doing so, the gut microbiota can colonise the pancreas, promoting tumorigenic inflammation. This could answer questions related to how transplanted microbiota can access the TME.

Indeed, if duodenal microbes translocate into the pancreatic tumour microbiome then duodenal fluid may serve as a microbial biomarker. Kohi et al. explored alterations in the duodenal fluid microbiome in patients with PDAC [55]. Their case control study compared bacterial and fungal (16S and 18S rRNA) profiles of secretin-stimulated duodenal fluid collections from 308 patients, 74 of which exhibited PDAC [55]. Patients with PDAC possessed a different duodenal fluid microbiome compared to pancreatic cysts and normal pancreata. There was an enrichment of *Bifidobacterium* in PDAC patients compared to controls. Furthermore, relative counts of *Fusobacteria, Rothia* and *Neisseria* were higher among patients with PDAC with short-term survival (<3.15 years) [55].

Langheinrich et al. conducted a prospective observational trial of 10 patients undergoing head of pancreas tumor surgery [38]. The aim of the study was to compare the microbiome at different body sites (bile duct, duodenal mucosa, pancreatic tumor lesion, postoperative drainage fluid and fecal samples). Their study demonstrates that there is a distinct microbiome in the various compartments adjacent to the pancreas. Patients that required pre-operative stenting possessed an altered microbiome with a higher amount of *Enterococcus*. The majority of patients with postoperative pancreatic fistula (POPF) showed a distinctly high abundance of *firmicutes* at the phylum level, predominantly represented by *Enterococcus*, found in bile and duodenal fluid samples. Comparing the texture of the pancreas gland, soft tissue contained *Fusobacteria*, while hard tissue consisted mainly of *Firmicutes, Proteobacteria and Verrucomicrobia*.

Belmouhand et al. investigated *Enterococci* (*E. faecium and E. faecalis*) obtained from drain samples to ascertain if *Enterococci* are associated with anastomotic leakage in patients undergoing pancreaticoduodenectomy (PD) [137]. *Enterococci* are known to degrade collagen [135]. Seventy patients were enrolled in the study, of whom 19 exhibited anastomotic leaks (18 pancreatic fistula and 1 hepaticojejunostomy leak). Patients with a leak carried a higher incidence of enterococci positive drain samples compared to those without a leak.

Species such as *P. Aeruginosa* and *Enterobacter cloacae* have been cultured from the drains of patients with pancreatic fistula after PD [156]. Such species produce increased amounts of proteases within the pancreas that activate trypsin [156]. This, in turn, may amplify tissue autolysis at the pancreaticojejunostomy site.

In combination, these studies suggest potentially clinically relevant alterations in the microbiome in tissues adjacent to PDAC. As previously discussed, these microbiotic changes may play a role in determining operative outcomes. However, further validation of the findings from these studies must be confirmed through larger scale studies and data sets.

Evidence to date suggests several routes by which microbiota can affect PDAC. There is likely a network of interactions between microbiota at anatomical specific sites that individually or collectively may protect or exacerbate the initiation or progression of disease in PDAC. The relationships between site specific microbiomes are dynamic in nature and influence each other to produce temporal changes with disease. Site specific microbial biomarkers in the future may serve in screening, strategizing therapeutics and prognostic aids.

#### 2.6.3. Bile Microbiota

Bactibilia is defined as microbes within bile. Maekawa et al. discovered the major microbes present in bile fluid of PDAC patients were *Enterobacter* and *Enterococcus* spp. [157]. They discovered *E. faecalis* in chronic pancreatitis and PDAC patients, as well as serum antibodies to *E. faecalis* in patients with chronic pancreatitis. They showed that pancreatic cells expressed pro-fibrotic cytokines when exposed to enterococcus lipoteichoic acid. Concluding infection with *E. faecalis* may facilitate the progression of chronic pancreatitis, ultimately leading to the development of PDAC. Serra et al. found that within head of pancreas tumours there was a negative correlation with *Escherichia coli* and *Pseudomonas* spp., whereas *Klebsiella* spp. was a positive predictor [158].

Liwinski et al. investigated alterations of the bile microbiome in primary sclerosing cholangitis (PSC) [159]. Fourty-three patients with PSC were compared to 22 control patients. A total of 260 biospecimens were obtained from the oral cavity, duodenum, mucosa and bile duct. Results firstly showed that the composition of bile duct microbiome is different from the other sites. Patients with PSC carried an increase level of *Enterococcus*, which is associated with lithocholic acid, a potentially carcinogenic bile acid [159].

Shrader et al. explored how contaminated bile effects survival of PDAC cells. Human PDAC cells were treated for 24 h with sterile (non-stented) bile, contaminated (stented) bile, and sterile bile pre-incubated with *Enterococcus faecalis* or *Streptococcus oralis*. Bile samples were obtained from patients undergoing a Pancreaticoduodenectomy. All sterile bile (*n* = 4) reduced PDAC cell survival in vitro, while contaminated bile samples exhibited a reduced anti-tumour effect [160]. Conversely, pre-incubation of sterile bile with live bacteria altered the antitumor effect of sterile bile [160]. This study showed that a change in the biliary microbiome through stenting directly impacts PDAC cell survival in vitro. Rogers et al. studied 50 patients who underwent pancreaticoduodenectomy, comparing bacterial communities (faecal, pancreatic, bile and jejunal microbiome profiles) and their clinical variables [27]. Comparison of the microbial communities showed that PDAC faecal samples were clustered together, clearly separated from bile, pancreas and jejunum samples. Similar communities were found within the pancreas, bile and jejunum of PDAC patients. We wonder whether bile microbial communities could be used as surrogate for pancreatic microbial communities in a bid to aid diagnosis and screening.

#### 2.6.4. Stool Microbiota

Considerable effort is being invested in investigating the stool microbiome in PDAC in a bid to tackle the poor survival rate of PDAC through early detection with biomarkers or a screening tool. However, work to date has fallen short of expectations.

Ren et al. compared the faecal microbiota of PDAC patients of Chinese origin and matched healthy controls [29]. Besides a lower alpha-diversity in patients with PDAC, a decreased abundance of Bifidobacterium species in the gut of PDAC patients compared to healthy controls, as well as a decreased abundance of some butyrate-producing bacteria (eg. *Coprococcus*, *Clostridium*, *Blautia*, *Flavonifractor* and *Anaerostipes* species) whilst an increased abundance of *Veillonella*, *Selenomonas* and *Klebsiella* species were found as well as lipopolysaccharide-producing bacteria (eg. *Prevotella*, *Hallella* and *Enterobacter* species). Whilst a PDAC faecal microbial signature has been demonstrated, the specificity and sensitivity of the signature was not strong enough to predict a low incidence disease such as PDAC [29]. A distinct faecal microbial signature has also been shown in an Israeli cohort and found to be distinct from healthy controls, as well as patients with liver damage or bile duct obstruction [161]. The specificity of their microbial classifiers was similar to that reported by Ren et al. [29]. Both Chinese and Israeli cohorts showed similar trends at the phylum level with an increase in *Bacteroidetes* and decrease in *firmicutes* in PDAC patients compared to controls. Whilst the Chinese and Israeli cohorts possessed shared phylum characteristics, they also possessed extensive differences in microbial composition. It is clear that variability in the microbiome exists not only at the host level, but also at a cohort level. Such variability may prove a hurdle to diagnostic testing and collaborative work.

Another aspect of the stool signature we must understand is the changes it may undergo during the course of disease and the functional implications of this. Two preclinical studies have longitudinally analysed the faecal microbiota profile in premalignant PDAC KC (Ptf1aCre, LSL-Kras^G12D^) mouse model compared to WT controls has already been discussed above [21,34].

#### 2.6.5. Cyst Fluid Biomarkers in IPMN

Among pancreatic cystic neoplasms (PCN), intraductal papillary mucinous neoplasms (IPMN) are the most common and are characterised by varying degrees of cellular dysplasia, from low-, to intermediate- and high-grade dysplasia and finally IPMN with an associated invasive carcinoma [162]. Whilst IPMN is often a precursor of PDAC, the preoperative prediction of IPMN is poor and the diagnosis is difficult without surgical intervention [66,163,164,165,166,167,168]. Given early pancreatic cystic lesions have been described to possess bacterial DNA, recent focus has turned to analysis of the microbiome in various anatomical sites, in search for a marker of IPMN [122,169]. Unfortunately, a pilot study investigating the oral microbiota for makers differentiating between controls, IPMN and PDAC did not demonstrate any positive results, with results generally similar between PDAC and IPMN cases [170].

A retrospective study using ultrasound fine needle aspiration to obtain pancreatic cyst fluid samples (*n* = 69), showed PCNs carry a unique microbiome that is independent of cyst type, clinical or biochemical parameters [169]. Leading on from this, Gaiser et al. prospectively investigated the intra-cystic microbiome in IPMN, assessing pancreatic resection tissue samples from patients suspected of having IPMN [122]. They found significantly elevated intra-cystic bacterial 16S DNA copy numbers in groups histologically classified as IPMN (10.1-fold higher mean, *p* = 0.0042) or invasive cancer (11.4-fold higher geometric mean, *p* = 0.0008) compared with normal samples. Furthermore, the median number of bacterial DNA counts as well as IL-1B levels were significantly higher in patients diagnosed with high grade dysplasia or cancer compared to low grade dysplasia.

What is of great interest is that this suggests intra-cystic pro-tumorigenic IL-1B levels may be driven by the presence of bacterial species. Indeed, the high levels of IL-1B found in high-grade IPMN lesions was not mirrored in preoperative IL-1B plasma levels. Furthermore, the levels of IL-1B in low-grade disease fluid samples were below the level of detection, whilst, in high-grade disease, it was significantly elevated. Whilst IL-1B’s expression in pancreatic cyst fluid may be a prognostic marker, with a positive predictive value and negative predictive value of of 71% and 75%, respectively, along with a sensitivity and specificity of 79% and 95%, respectively; the exact role of IL-1B in promoting or inhibiting histological progression and interaction with bacteria in PDAC remains to be determined [171].

Furthermore, 15 bacterial genera have been found to be differentially abundant dependent on lesion classification as low-grade, high-grade or cancer, and the identification of specific bacterial markers, such as *F. nucleatum* in the cyst fluid of IPMNs, may be indicative of high-grade disease [122]. It may be that specific bacteria are found to be “driver” species leading to carcinoma or simply may serve as a biomarker to discriminate or stratify disease severity prior to surgical intervention or provide a means for surveillance in patients in whom reoccurrence following resection is possible or progression in conservatively managed patients is possible. This is, of course, dependent on validation in larger populations and case control studies. For now, the exact role bacteria play in IPMN is intriguing, yet not clear.

#### 2.6.6. Role of Bacterial Extracellular Vesicles in PDAC

Extracellular vesicles (EVs) are nano-sized membrane-bound vesicles released from cells, capable of transporting cargo. Cargo can include nucleic acids, proteins, lipids, amino acids, and metabolites. All cells, prokaryotic and eukaryotic release extracellular vesicles, with their significance a source of ongoing exploration. So far, they have been identified to have roles in immune responses, viral pathogenicity, pregnancy, cardiovascular diseases, central nervous system related diseases, and cancer progression [172].

BEVs are produced in several ways. Gram negative bacteria produce their vesicles via one of two main pathways. The first, and archetypal, method involves outer membrane blebbing, produced by disruption of the crosslinks between outer membrane and the peptidoglycan cell wall layer, creating outer membrane vesicles. The second is a result of explosive cell lysis, giving outer-inner membrane vehicles or explosive outer membrane vehicles. Gram positive bacteria, meanwhile, produce cytoplasmic membrane vesicles via endolysin-triggered bubbling cell death, giving cytoplasmic membrane vehicles [24,173,174,175].

Increasingly it is apparent that BEVs provide a means of inter-kingdom communication, allowing host and microbiome to interact and influence one another. BEVs exert immunomodulatory effects, with the ability to dampen, enhance or exaggerate the immune response. BEVs can bear ligands on their surface, such as LPS, LTA and peptidoglycan, all of which can be recognised by TLRs, including TLR2 and 4 on the host cell surface. Meanwhile, BEVs can also be internalised via lipid-raft dependent and independent endocytosis, as well as dynamin, caveolin and clarithin dependent entry. The intracellular compartment can detect bacterial cargo introduced, leading to a milieu of different responses. For example, through TLRs 3, 7, 8 and 12, bacterial RNA cargo can be sensed intracellularly. Both intracellular and extracellular BEV interactions can lead to the generation of an immune response, with the associated production of cytokines, chemokines, co-stimulatory molecules, and immune cell recruitment [24,174,175,176,177,178,179,180,181,182,183].

BEVs have been known to traverse across the gut epithelial barrier, allowing them to enter the submucosa (Figure 6). However, BEVs may hold a reach far beyond the gut. More recently, studies have looked at the ability of BEVs to enter the circulation. In a study comparing a group of healthy patients to those with intestinal barrier dysfunction (colitis, cancer radiation, chemotherapy induced colitis and treatment naïve HIV), higher levels of BEVs were found in the circulation in those with intestinal barrier dysfunction. They also found an increased plasma zonulin levels (a biomarker of barrier integrity) correlated with BEV levels, inferring associations between barrier dysfunction, and circulating BEVs [184]. Furthering their work, the same group showed the ability to isolate BEVs in various tissues, using a novel method relying on the densities of BEVs, utilising ultrafiltration, size exclusion chromatography and density-gradient centrifugation. This novel protocol allows for the isolation of BEVs from various tissues within 72 h and retains the BEVs structure and cargo. It also identifies the relative abundance of gram negative and positive BEVs, ultrastructure and function in a variety of tissues. In their study, they were able to show the presence of BEVs in the blood plasma of non-septic patients with concentrations ranging from 10^5^ to 10^6^. This protocol was only developed 18 months ago; we anticipate the field of BEVs will expand rapidly as the contribution of BEVs to various disease states are explored [175].

Whilst the presence of BEVs in healthy controls may seem surprising, it has been demonstrated that BEVs in the gut can access systemic tissues within hours of ingestion. In a mouse model, 8 h after oral ingestion of far-red fluorescent DiD-labelled *Bacteriodes thetaiotaomicron* outer membrane vesicles, non-GI organs showed staining. This included the heart and lungs, but was most intense in the liver. This suggests that gut BEVs can enter the portal circulation and the systemic circulation and can reach distant organs. It can be imagined that BEVs can act as long-distance messengers from the gut, communicating with the host organs in distant sites. However, how BEVs impact normal physiology or if they can influence the hallmarks of cancer in PDAC is yet to be determined. However, given the rich portal and systemic circulation encompassing the pancreas and specifically head of pancreas, it is logical to consider the role gut BEVs may serve in PDAC.

Indeed, Kim et al. have more recently looked at the use of serum BEVs in search for a marker for PDAC. Utilising 16rRNA analysis, they determined if any differences could be found between PDAC patients (*n* = 38) and healthy controls (*n* = 52) [185]. Across 90 samples, they identified between group differences at the phylum and genus levels. Interestingly, they found an increased alpha diversity in the PDAC group at the phylum level, but no difference between groups at the genus level. Beta diversity (between sample diversity) was found between the two groups to be significant, with principal coordinate analysis demonstrating distinction between PDAC and control groups at both phylum and genus level. Most significantly, however, their study only used 16rRNA bacterial genomic data from blood samples.

Utilising this data, they produced a prediction model, considering all possible combinations, utilising randomly separated model development. The bacterial markers they found to produce the best models were *Verrucomicrobia* and *Actinobacteria* at the phylum level. This model showed a perfect sensitivity of 1.0 in both the training set and the cross-validation set, with specificity of 0.846 in both testing and validation and training ROC AUC of 0.966 and 0.962 when cross validated. The ideal model utilised microbiome markers from the geneses of *Sphingomonas*, *Ruminoccaceae UCG-014*, *Propionibacterium*, *Akkermansia*, *Ruminiclostridium*, *Lachnospiraceae UCG-001* and *Corynebacterium 1*. This model also showed perfect sensitivity and ROC AUC in training and cross-validation cohorts with a specificity of 0.8462 and 0.9231 in training and cross-validation, respectively. This study shows the ability to utilise the genomic data that is held within BEVs in the circulation to identify the microbiomic changes. These can be associated with PDAC and, thus, larger studies investigating whether PDAC can be diagnosed from peripheral blood samples, which look for microbiomic signatures, not tumour produced or related proteins, cell free DNA or metabolites as conventional lines of investigation previously pursued, are urgently needed [185,186].

Crucially, this group also sought to determine the biological function of BEVs. Utilising the BEVs isolated from C. glutamicum, which was almost the same as that from an identified biomarker bacterium *Corynebacterium 1*, they looked at the impact on TNF-alpha production. They demonstrated an ability to suppress the TNF-alpha production in a dose dependent manner and suggest that this may be significant, as *Corynebacterium* was identified as significantly reduced in PDAC patients. The authors suggest, based on these in vitro findings, that the absence of this genus may provide an inflammatory environment permitting tumour development [185].

It is apparent that, as we begin to appreciate the microbiome and its impact on gut health as well as wider human health, how the microbiome exerts these effects is also under investigation. BEVs entering the circulation and accessing distant organs is one means by which this occurs. Crucially, the genetic cargo of BEVs can give an insight into the composition of the microbiome via peripheral blood samples. As we begin to understand how BEVs may impact the organs that they arrive at, it may be that we no longer consider an isolated gut microbiome, or tumour microbiome, but a complex and dynamic super-organism, composed of various microbial cells and human somatic cells in a constant state of cross talk and feedback via distant messenger vehicles such as BEVs. This, however, remains hypothetical at this present juncture, with further investigation of BEVs in the next few years likely crucial to the development of this theory.

## 3. Conclusions

Microbial communities are in constant communication and states of feedback between site specific microbial communities within the host. This state of communication can have implications on the local tumour microbiome, as well as distant microbiomes; they are all likely influenced by the host’s environmental factors.

In PDAC, it is apparent that the local tumour microbiome and distant anatomical specific microbiomes may impact PDAC progression and outcomes. Continuing to develop our understanding of this dynamic dialogue is imperative to our understanding of PDAC and to improving outcomes. We must consider the host as an entire ecosystem with changes in one anatomical specific site causing ripple effects throughout the host as a whole. Understanding the microbiome is important, but so is contextualising the information. With the microbiome comes the complexity of site and source of communities being examined, as well as the wider context of the patient’s health status. The tumour microbiome provides some information, and the gut provides yet more, with the significance of bile and other site-specific microbiomes currently being investigated. As we develop our understanding of these -omics based fields, we will also need to develop how we link these (i.e., how they communicate and influence shifts in the make-up of any given microbiome) and what they mean when integrated. It may be that the information gleaned from the tumour and gut microbiome composition in concert with the tumour genome provides a level of resolution that allows for the assessment of disease status and prognosis.

Indeed, there is much room to develop a comprehensive understanding of the microbiota and the mechanisms of inducing therapeutic responses. Through such understanding, we can advance therapeutic options, as well as identify biproducts of pathways that may be used as biomarkers of disease with a role in diagnostics, surveillance and predicting or monitoring treatment response. Further steps in developing this field would be in silico-based studies, collating the UGI tract, LGI tract, biliary tree and tumour microbial composition and the tumour genome, assessing for prognostically favourable outcomes. Other steps include proving a causative role for microbes within the TME in PDAC, as opposed to an associative role. It may be that the interaction between the microbiome and tumours extends across numerous hallmarks of cancer. It can be imagined that different cancers may possess different “driver” microbes and, equally, that tumour environments may select for certain species and microbial community compositions. With this in mind, as we develop our understanding, the microbiome may gain further recognition as a hallmark of cancer in its own right or, alternatively, at the centre of this framework, with a pivotal role in stimulating or opposing the hallmarks of cancer. Either way, becoming part of the framework that conceptualises the progression from normal tissue to cancer. From this information, we may be able to develop new insights into favourable therapeutic targets in both the microbiome and tumour genome. In doing so, we may be able to develop an anatomical map of microbiota significance in order to advance the application of diagnostic and therapeutic strategies that may enhance PDAC prognosis.

As discussed above, evidence suggests that bacterial signatures may be employed to differentiate between cancer and non-cancer phenotypes, as well as between cancer type and stage in certain circumstances. To date, no study has been able to highlight a specific microbial biomarker with high sensitivity and specificity that could serve in screening or diagnostics; it may be that the future of microbial biomarkers in screening lies in targeting individuals at risk of PDAC or those with pre-cancerous pancreatic neoplasms. Or it could be that microbial biomarkers must be used in combination with alternative biomarkers in order to improve sensitivity and specificity.

Recent advances in advanced sequencing methods have facilitated our understanding of the microbiota associated with PDAC TME. This technology, in combination with research initiatives, should revolutionise the study of human subjects. Several human trials are currently underway to explore the therapeutic potential for microbiome modulation in PDAC. Whilst we hope to gain more information through these trials, we must remain aware that the microbiome of the host is likely a super-organism in a constant state of attempted balance and communication through various conceptual axis which have systemic influences that may alter the balance between distant site-specific microbiomes. We should aim to capture this systemic state of play in future studies and create a road map of the body’s anatomical microbiota to guide future work, as the imbalance invoked by microbiome modulation may result in subsequent undesirable disease states. Studies should have an appreciation for the collateral damage that may ensue. While a large amount of our understanding is derived from animal models, the lack of progress in the field of PDAC suggests a need for bold rethinking when considering PDAC therapy and the need to make a start in applying these in human trials.

## Figures and Tables

**Figure 1 cancers-14-01020-f001:**
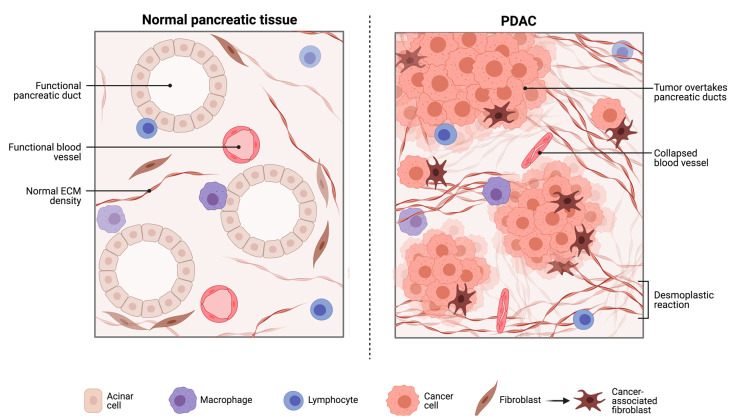
The PDAC TME is heterogenous and poorly characterised. The PDAC TME contains tumour associated macrophages (TAM), myeloid-derived suppressor cells (MDSCs) and regulatory T-cells (T-regs) that are all involved in immunosuppressive tumour promoting activity. Furthermore, there are dense desmoplastic reactions as well as collapsed vessels which all provide barriers to cytotoxic T cell infiltration targeting PDAC tumour cells. The TME affords PDAC protection against chemo- and immune-therapeutics.

**Figure 2 cancers-14-01020-f002:**
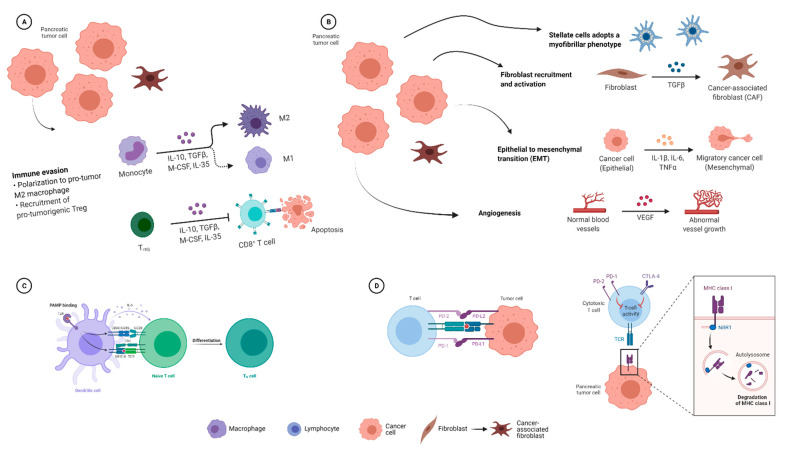
Overview of the immunosuppression in PDAC, associated with the presence of tumour-promoting immune cells. (**A**) Pancreatic tumour cells, produce an immune phenotype rich with an increase in infiltrates FOXP3, Tregs and M2 polarised macrophages that lead to poorer prognosis. (**B**) Pancreatic cancer cells promote angiogenesis and epithelial to mesenchymal transition (EMT). EMT is loss of epithelial e-cadherin and increase in mesenchymal Vimentin allows cancer cells to become more mobile and metastatic. Stellate cells are responsible for the profound desmoplasia observed in PDAC. Cancer-associated fibroblasts (CAFs) aid tumour growth, local invasion and metastasis. (**C**) Dendritic cells (DC) are antigen presenting cells (APCs) that generate tumour-protective T cells within the TME of PDAC. (**D**) One of the strategies used by PDAC is to bypass the immune surveillance by the misuse of immune checkpoints in order to escape immune recognition. Cytotoxic T lymphocyte-associated antigen 4 (CTLA-4 or CD152) and programmed cell death protein 1 (PD-1 or CD279) are co-inhibitory receptors of T cell receptor (TCR) signalling. Immune checkpoints inhibit T-cell activation. PD-L1 has been reported to be overexpressed in PDACs, and this overexpression correlates with worse prognosis of the patients.

**Figure 3 cancers-14-01020-f003:**
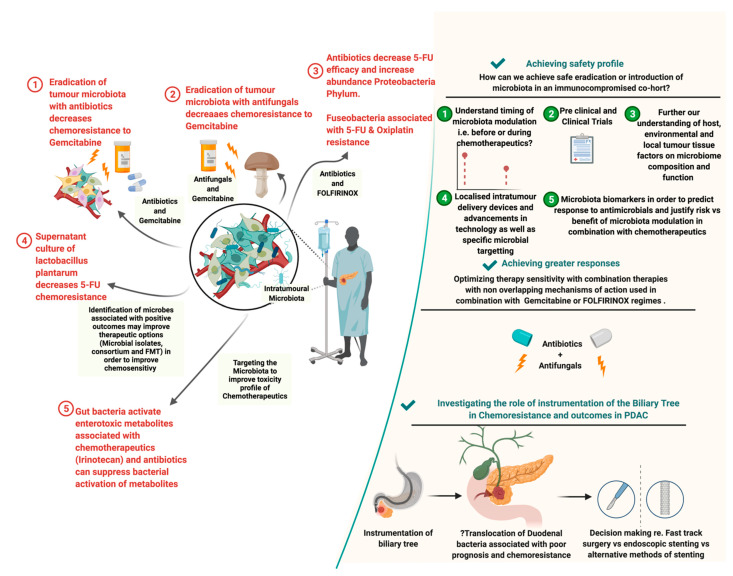
Highlighting (**left**) current findings and the potential roles microbiome modulation may play in the future of PDAC chemotherapeutics. In this illustration (**right**), we also suggest several focuses of future research which may prove critical in establishing the role for microbiome modulation and chemotherapeutics.

**Figure 4 cancers-14-01020-f004:**
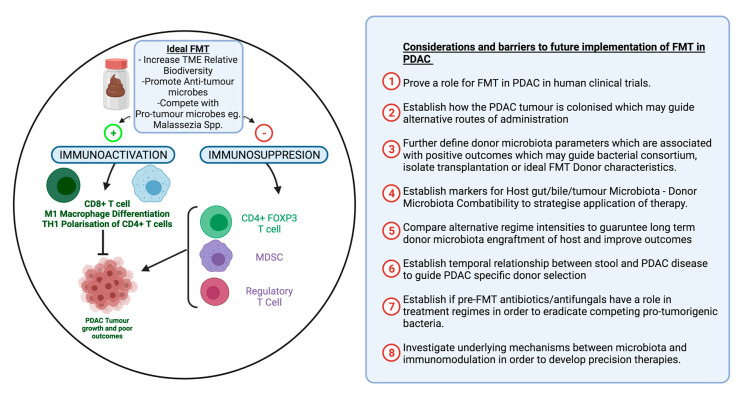
The ideal FMT must have a composition which promotes immunoactivation and inhibits immunosuppression within the TME (**left**). Current considerations and barriers to FMT implementation in PDAC are highlighted (**right**).

**Figure 5 cancers-14-01020-f005:**
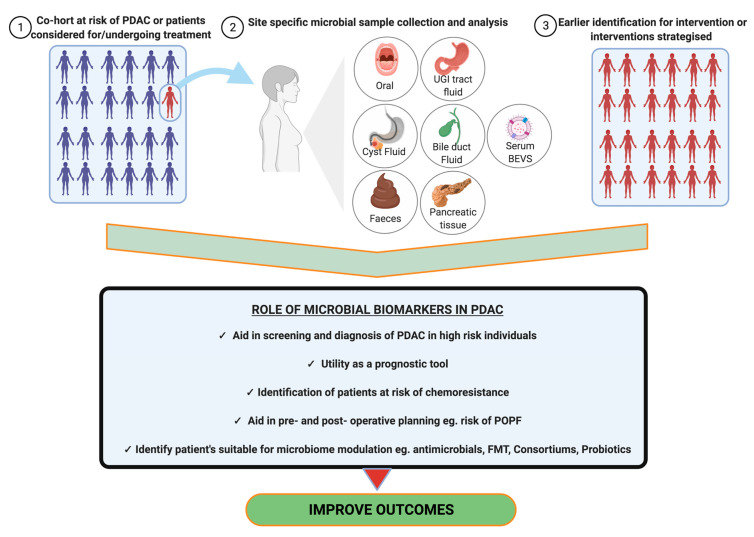
Conceptual application of microbial biomarkers in PDAC. In at risk cohorts, an array of site-specific sampling (non-invasive vs invasive) may provide an avenue for early detection of PDAC in at risk individuals, which may serve to identify more individuals suitable for intervention or alternatively may serve as a means for predicting outcomes of their disease and subsequent oncological therapies. Site specific microbial biomarkers or serum-based assays of bacterial endovesicles (BEVs) may be used in combination or in isolation at various stages in the patients journey with applications throughout the spectrum beginning at initial screening/diagnosis and spanning across to post-treatment surveillance.

**Figure 6 cancers-14-01020-f006:**
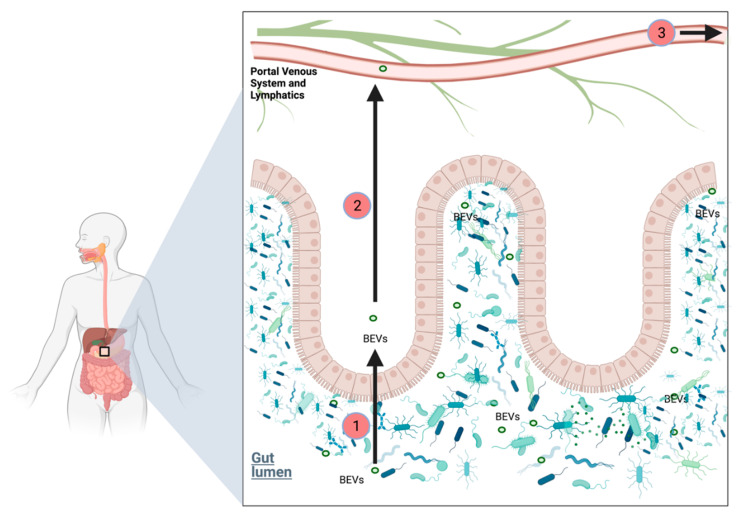
Bacterial extracellular vesicles produced by gut bacteria cross the gut epithelial barrier to the submucosa via transcellular and/or paracellular transport, exacerbated by disruption of the gut wall barrier (1). They subsequently enter the portal circulation and/or lymphatics (2) and in doing so may provide a means for inter-kingdom communication with site specific microbiomes such as the PDAC microbiome (3). The resultant affect may be one of immunomodulation locally at the tumour microbiome. BEVs in the circulation may be used as biomarkers of PDAC.

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
