# Peer review of "A Comprehensive Review of the Current and Future Role of the Microbiome in Pancreatic Ductal Adenocarcinoma"

_cancers, 2022, doi:10.3390/cancers14041020_

Round 1
Reviewer 1 Report
The manuscript discusses the possible influence of the microbiome on the prognosis and outcome of pancreatic cancer. The microbiome has become a subject of interest in the last ten years. In particular the relationship microbiome-cancer has been explored extensively and there are important publications in this regard. None of them are mentioned by the authors. I suggest to include references such as
Bhatt, A. P., Redinbo, M. R., & Bultman, S. J. (2017). The role of the microbiome in cancer development and therapy. CA: a cancer journal for clinicians, 67(4), 326-344.
Picardo, S. L., Coburn, B., & Hansen, A. R. (2019). The microbiome and cancer for clinicians. Critical reviews in oncology/hematology, 141, 1-12.
Gopalakrishnan, V., Helmink, B. A., Spencer, C. N., Reuben, A., & Wargo, J. A. (2018). The influence of the gut microbiome on cancer, immunity, and cancer immunotherapy. Cancer cell, 33(4), 570-580.
Whisner, C. M., & Aktipis, C. A. (2019). The role of the microbiome in cancer initiation and progression: how microbes and cancer cells utilize excess energy and promote one another’s growth. Current nutrition reports, 8(1), 42-51.
The style adopted by the authors runs out of focus frequently and without justification. For example: they discuss the Hallmarks by Hanahan and Weinberg listing all of them. Actually the microbiome, which is the central issue of this paper is not mentioned even once in the two publications of the Hallmarks.
The authors bring up many highly speculative ideas without references to sustain them. For example in the following sentence:
“Of interest would be the introduction of tumour-based vaccines against epitopes expressed by the microbiome. These therapies would target more than just the tumour, but also the tumour supporting microenvironment. Key questions would be whether this would change the efficacy of vaccines, and how vaccines targeting antigens on PDAC and specific taxa associated with STS impacts outcomes, compare with vaccines targeting PDAC antigens alone.”
Similar speculation happens in the following paragraph:
“We propose the microbiome contributes to survival differences via the complex interaction of various pathways. Invasion of the pancreas by certain species may generate certain metabolites in the tumour, generating signals within the TME favouring a particular cancer phenotype through immune pathways (Figure 1)”.
This needs at least a reference.
Regarding microbiota and immunology, important references are missing such as
Fessler, J., Matson, V., & Gajewski, T. F. (2019). Exploring the emerging role of the microbiome in cancer immunotherapy. Journal for immunotherapy of cancer, 7(1), 1-15.
Li, W., Deng, Y., Chu, Q., & Zhang, P. (2019). Gut microbiome and cancer immunotherapy. Cancer letters, 447, 41-47.
Schwartz, D. J., Rebeck, O. N., & Dantas, G. (2019). Complex interactions between the microbiome and cancer immune therapy. Critical reviews in clinical laboratory sciences, 56(8), 567-585.
Armstrong, H., Bording-Jorgensen, M., Dijk, S., & Wine, E. (2018). The complex interplay between chronic inflammation, the microbiome, and cancer: understanding disease progression and what we can do to prevent it. Cancers, 10(3), 83.
Lee, K. A., Shaw, H. M., Bataille, V., Nathan, P., & Spector, T. D. (2020). Role of the gut microbiome for cancer patients receiving immunotherapy: Dietary and treatment implications. European Journal of Cancer, 138, 149-155.
An important reference that cannot be excluded and discussed is
McQuade, J. L., Daniel, C. R., Helmink, B. A., & Wargo, J. A. (2019). Modulating the microbiome to improve therapeutic response in cancer. The Lancet Oncology, 20(2), e77-e91.
When they discuss gammaproteobacteria and resistance to gemcitabine the following references should be cited and /or discussed
Choy, A. T., Carnevale, I., Coppola, S., Meijer, L. L., Kazemier, G., Zaura, E., ... & Giovannetti, E. (2018). The microbiome of pancreatic cancer: from molecular diagnostics to new therapeutic approaches to overcome chemoresistance caused by metabolic inactivation of gemcitabine. Expert review of molecular diagnostics, 18(12), 1005-1009.
Thomas, H. (2017). Intra-tumour bacteria promote gemcitabine resistance in pancreatic adenocarcinoma. Nature Reviews Gastroenterology & Hepatology, 14(11), 632-632.
McAllister, F., Khan, M. A. W., Helmink, B., & Wargo, J. A. (2019). The tumor microbiome in pancreatic cancer: bacteria and beyond. Cancer cell, 36(6), 577-579.
Other highly speculative paragraph with no references is the following:
“The microbiome may produce factors that either inhibit CD8+ T-cells or induce a Th2/Treg rich immune phenotype, thereby generating a tumour-permissive environment. Equally, in LTS, the microbiome may provide certain neoantigens that allow for the molecular mimicry of tumour antigens, leading to immune activation, in a similar manner to certain forms of autoimmune disease, thus resulting in cancer immune surveillance. Finally, there may be an aspect by which during tumourigenesis, tumour promoting mutations occur in a distinct sequence, which creates a microenvironment favouring pancreatic colonisation by certain taxa, which are in turn noted to correlate with LTS. These infect the tumour and establish a tumour permissive or regulating niche within the TME. However, this would go against the ability to modulate survival by altering the microbiome (e.g. by faecal microbial transplant or antibiotics), as discussed later. We believe that a dynamic combination of the aforementioned processes likely occur within the tumour-microbiome interface with crosstalk impacting the immune response. We postulate that the microbiome in STS may contribute to cancer hallmarks through modulation of the metabolome resulting in shorter survival in these patients, namely deregulation of cellular energetics, avoiding immune destruction, and tumour promoting inflammation”
Authors say: ” One key to unlocking the efficacy of immunotherapy in PDAC may be found in treating early-stage PDAC, when peripancreatic inflammation promoted by microbiomic oncogenic signalling and suppression of the innate and adaptive immune response is not yet fully established”
This is also speculative. Unless it can be supported by a reference, it has not been appropriately proved that microbiome generates oncogenic signaling. Peripancreatic inflammation can also be a product of enzymes leakage from the tumor and cytokines produced by stellate cells.
Authors say “It can be imagined that PDAC microbiome may provide an accessible source of nutrients to sustain growth under the correct conditions.”
Again, this is speculative unless they can provide an adequate reference.
Figure 2: stellate cells should be shown and mentioned. Cancer associated fibroblasts and stellate cells are different. While the last adopts a myofibrillar phenotype, the first do not.
Authors say “This is significant as PDAC is recognised to grow in a nutrient poor environment, and it has been shown to overcome this by actively scavenging extracellular proteins for growth via micropinocytosis [62]”
They should add that micropinocytosis is only one of the mechanisms to overcome lack of nutrients. The other important mechanism used by PDAC is adopting an autophagic phenotype.
This sentence needs to be improved. May be make it shorter?
“This review examines the role of the microbiome in PDAC, assessing how it may alter survival outcomes, postulating mechanisms by which it does this, evaluating the possibility of employing microbiomic signatures as biomarkers of PDAC and examining whether it may be amenable to targeting, altering it and consequently & a word is missing here & the natural history of PDAC.”
Reference 6 is not backing what is said in the paragraph. I suggest to remove it.
The following paragraph needs a reference: “More recently, the microbiome has been recognised as having a pivotal role in influencing the immune system, interactions with cancer therapeutics and outcomes in PDAC.”
Typo in line 147.
Line 149. Explain what alpha-diversity means. For example: as the mean diversity of species in different sites or habitats within a local scale.
Line 210 says “Interestingly, the depletion of CD8+T-cells in these mice led to the complete loss of anti-tumour response, while only partial loss was observed with CD4+ T-cell depletion [40].”
There is nothing interesting about these findings. This is what is expected to logically happen. I suggest removing the word “Interestingly”.
Author Response
The manuscript discusses the possible influence of the microbiome on the prognosis and outcome of pancreatic cancer. The microbiome has become a subject of interest in the last ten years. In particular the relationship microbiome-cancer has been explored extensively and there are important publications in this regard. None of them are mentioned by the authors. I suggest to include references such as
Bhatt, A. P., Redinbo, M. R., & Bultman, S. J. (2017). The role of the microbiome in cancer development and therapy. CA: a cancer journal for clinicians, 67(4), 326-344.
Reference included
Picardo, S. L., Coburn, B., & Hansen, A. R. (2019). The microbiome and cancer for clinicians. Critical reviews in oncology/hematology, 141, 1-12.
Reference included
Gopalakrishnan, V., Helmink, B. A., Spencer, C. N., Reuben, A., & Wargo, J. A. (2018). The influence of the gut microbiome on cancer, immunity, and cancer immunotherapy. Cancer cell, 33(4), 570-580.
Reference included
Whisner, C. M., & Aktipis, C. A. (2019). The role of the microbiome in cancer initiation and progression: how microbes and cancer cells utilize excess energy and promote one another’s growth. Current nutrition reports, 8(1), 42-51.
Reference included
The style adopted by the authors runs out of focus frequently and without justification. For example: they discuss the Hallmarks by Hanahan and Weinberg listing all of them. Actually the microbiome, which is the central issue of this paper is not mentioned even once in the two publications of the Hallmarks.
This paragraph has been modified to reflect the reviewer’s comments. Additionally, Hanahan has since published a further update to the hallmarks framework, recognising the microbiome as an enabling characteristic. This has been cited accordingly.
Hallmarks of Cancer: New Dimensions, Douglas Hanahan, Cancer Discov January 1 2022 (12) (1) 31-46; DOI: 10.1158/2159-8290.CD-21-1059
The authors bring up many highly speculative ideas without references to sustain them. For example in the following sentence:
“Of interest would be the introduction of tumour-based vaccines against epitopes expressed by the microbiome. These therapies would target more than just the tumour, but also the tumour supporting microenvironment. Key questions would be whether this would change the efficacy of vaccines, and how vaccines targeting antigens on PDAC and specific taxa associated with STS impacts outcomes, compare with vaccines targeting PDAC antigens alone.”
This has been removed to reflect the reviewers comments.
Similar speculation happens in the following paragraph:
“We propose the microbiome contributes to survival differences via the complex interaction of various pathways. Invasion of the pancreas by certain species may generate certain metabolites in the tumour, generating signals within the TME favouring a particular cancer phenotype through immune pathways (Figure 1)”.
This needs at least a reference.
We appreciate the reviewer’s input on this. Reference has now been made to a study previously cited in the body of the work, as well as Hanahan’s latest update to the hallmarks of cancer and numerous other studies. We appreciate that this portion of the work remains speculative, however we highlight it as an avenue of exploration, with evidence of similar models in other cancers now provided.
Regarding microbiota and immunology, important references are missing such as
Fessler, J., Matson, V., & Gajewski, T. F. (2019). Exploring the emerging role of the microbiome in cancer immunotherapy. Journal for immunotherapy of cancer, 7(1), 1-15.
This review article has not been included. With regards to PDAC, this review discusses papers we have already discussed in text.
Li, W., Deng, Y., Chu, Q., & Zhang, P. (2019). Gut microbiome and cancer immunotherapy. Cancer letters, 447, 41-47.
This letter has not been included. The letter does not specifically discuss PDAC and may be beyond the scope of our review.
Schwartz, D. J., Rebeck, O. N., & Dantas, G. (2019). Complex interactions between the microbiome and cancer immune therapy. Critical reviews in clinical laboratory sciences, 56(8), 567-585.
This review article has not been included. With regards to PDAC, this review discusses papers we have already discussed in text.
Armstrong, H., Bording-Jorgensen, M., Dijk, S., & Wine, E. (2018). The complex interplay between chronic inflammation, the microbiome, and cancer: understanding disease progression and what we can do to prevent it. Cancers, 10(3), 83.
Referenced in text.
Lee, K. A., Shaw, H. M., Bataille, V., Nathan, P., & Spector, T. D. (2020). Role of the gut microbiome for cancer patients receiving immunotherapy: Dietary and treatment implications. European Journal of Cancer, 138, 149-155.
Referenced in text
An important reference that cannot be excluded and discussed is
McQuade, J. L., Daniel, C. R., Helmink, B. A., & Wargo, J. A. (2019). Modulating the microbiome to improve therapeutic response in cancer. The Lancet Oncology, 20(2), e77-e91.
This reference has now been included.
When they discuss gammaproteobacteria and resistance to gemcitabine the following references should be cited and /or discussed
Choy, A. T., Carnevale, I., Coppola, S., Meijer, L. L., Kazemier, G., Zaura, E., ... & Giovannetti, E. (2018). The microbiome of pancreatic cancer: from molecular diagnostics to new therapeutic approaches to overcome chemoresistance caused by metabolic inactivation of gemcitabine. Expert review of molecular diagnostics, 18(12), 1005-1009.
We have referenced this paper and discussed their opinions/conclusions.
Thomas, H. (2017). Intra-tumour bacteria promote gemcitabine resistance in pancreatic adenocarcinoma. Nature Reviews Gastroenterology & Hepatology, 14(11), 632-632.
This reference has not been included. We do not feel this reference adds anything to our work as they simply explain in brief the work carried out by Geller and colleagues (2017) and add no opinions or conclusions of their own based on the work conducted. We have discussed Geller and colleagues work in our review.
McAllister, F., Khan, M. A. W., Helmink, B., & Wargo, J. A. (2019). The tumor microbiome in pancreatic cancer: bacteria and beyond. Cancer cell, 36(6), 577-579.
Cited.
Other highly speculative paragraph with no references is the following:
“The microbiome may produce factors that either inhibit CD8+ T-cells or induce a Th2/Treg rich immune phenotype, thereby generating a tumour-permissive environment. Equally, in LTS, the microbiome may provide certain neoantigens that allow for the molecular mimicry of tumour antigens, leading to immune activation, in a similar manner to certain forms of autoimmune disease, thus resulting in cancer immune surveillance. Finally, there may be an aspect by which during tumourigenesis, tumour promoting mutations occur in a distinct sequence, which creates a microenvironment favouring pancreatic colonisation by certain taxa, which are in turn noted to correlate with LTS. These infect the tumour and establish a tumour permissive or regulating niche within the TME. However, this would go against the ability to modulate survival by altering the microbiome (e.g. by faecal microbial transplant or antibiotics), as discussed later. We believe that a dynamic combination of the aforementioned processes likely occur within the tumour-microbiome interface with crosstalk impacting the immune response. We postulate that the microbiome in STS may contribute to cancer hallmarks through modulation of the metabolome resulting in shorter survival in these patients, namely deregulation of cellular energetics, avoiding immune destruction, and tumour promoting inflammation”
This has now been referenced and modified.
Authors say: ” One key to unlocking the efficacy of immunotherapy in PDAC may be found in treating early-stage PDAC, when peripancreatic inflammation promoted by microbiomic oncogenic signalling and suppression of the innate and adaptive immune response is not yet fully established”
This is also speculative. Unless it can be supported by a reference, it has not been appropriately proved that microbiome generates oncogenic signaling. Peripancreatic inflammation can also be a product of enzymes leakage from the tumor and cytokines produced by stellate cells.
Many thanks for your feedback. This line has now been removed.
Authors say “It can be imagined that PDAC microbiome may provide an accessible source of nutrients to sustain growth under the correct conditions.”
Again, this is speculative unless they can provide an adequate reference.
This now reads as: “Studies are required on the contribution (if any) of PDAC microbiomic products as a source of nutrients to sustain PDAC growth.”
Figure 2: stellate cells should be shown and mentioned. Cancer associated fibroblasts and stellate cells are different. While the last adopts a myofibrillar phenotype, the first do not.
Figure updated and stella cells added.
Authors say “This is significant as PDAC is recognised to grow in a nutrient poor environment, and it has been shown to overcome this by actively scavenging extracellular proteins for growth via micropinocytosis [62]”
They should add that micropinocytosis is only one of the mechanisms to overcome lack of nutrients. The other important mechanism used by PDAC is adopting an autophagic phenotype.
Added text as suggested.
This sentence needs to be improved. May be make it shorter?
“This review examines the role of the microbiome in PDAC, assessing how it may alter survival outcomes, postulating mechanisms by which it does this, evaluating the possibility of employing microbiomic signatures as biomarkers of PDAC and examining whether it may be amenable to targeting, altering it and consequently & a word is missing here & the natural history of PDAC.”
Paragraph now shortened.
Reference 6 is not backing what is said in the paragraph. I suggest to remove it.
Reference now removed.
The following paragraph needs a reference: “More recently, the microbiome has been recognised as having a pivotal role in influencing the immune system, interactions with cancer therapeutics and outcomes in PDAC.”
Typo in line 147.
Typo addressed.
Line 149. Explain what alpha-diversity means. For example: as the mean diversity of species in different sites or habitats within a local scale.
Explained in text.
Line 210 says “Interestingly, the depletion of CD8+T-cells in these mice led to the complete loss of anti-tumour response, while only partial loss was observed with CD4+ T-cell depletion [40].”
There is nothing interesting about these findings. This is what is expected to logically happen. I suggest removing the word “Interestingly”.
Interestingly has now been removed.
Reviewer 2 Report
The authors provide a comprehensive review of microbiome in PDAC and how it contributes to disease outcome and therapeutic outcome.
The review doesn't address PDAC subtypes. PDAC is composed of sub types (classical, basal) and it has been well studied that certain subtypes have poor survival and treatment response. The authors need to address the subtypes and if there is any correlation published with microbiome and subtypes.
Author Response
Many thanks for your feedback. We have now addressed PDAC subtypes and discussed a paper that has investigated the association between PDAC subtypes and the tumour microbiome.
Round 2
Reviewer 1 Report
Line 68 Some have proposed…
It should say Some authors have proposed
Line 103 Previously, work has confirmed
Revise syntax. It should be Previous works have confirmed or Previous research has confirmed
Line 153 PDAC, however, is not as immunogenic.
Revise syntax.
Line 320 Epithelial
No capital letter to be used..
Line 493 quantities of <5%
Please change it for quantities of less than 5%
Line 497 This positive effect of FMT with LTS material was then ablated with antibiotics
Please change the word ablated for some other word such as eliminated, lost, impeded, or any other,
Line 516 However, results were not reproducible for other fungal genera such as
Revise syntax
Line 634 it is well regarded
Please change it for it is well known
Line 641 it is postulated
Change verb tense for has been postulated
And please revise the syntax of the whole sentence.
Line 660 . The Figure
Eliminate The
Line 673 . Statistically significant between group differences were identified
Replace for: Statiscally significant differences between groups were identified.
Line 684 Pushalkar et al. investigated the possibility antibiotics could impact
Add that. Pushalkar et al. investigated the possibility that antibiotics could impact
Line 699 This is essential to targeting the tumour promoting and sustaining aspects of the microbiome,
Revise syntax. The sentence is not clear.
Line 704 Doing as such may end up
Revise syntax. Probably the authors meant: Doing that, may end up.
Line 721: 2.5. Modulation in perioperative period subsection
Please change it for 2.5. Modulation in the perioperative period
Eliminate subsection.
Line 820 Change who by its
Author Response
Many thanks for your feedback. All grammatical changes have been made as suggested.